# SAGA: Structural Aggregation Guided Alignment with Dynamic View and Neighborhood Order Selection for Multiview Graph Domain Adaptation

**Ruiyi Fang**[1]  **Jingyu Zhao**[1]  **Shuo Wang**[2]  **Ruizhi Pu**[1]  **Bingheng Li**[3]  **Jiale Cai**[1]
**Zhihao Li**[1]  **Zihao Jing**[1]  **Zhu Jian**[4]  **Song Tang**[5]  **Charles Ling**[1]  **Boyu Wang**[1,6*]
[1] Western University      [2] University of Electronic Science and Technology of China
[3] Michigan State University      [4] Guangdong University of Technology
[5] University of Shanghai for Science and Technology      [6] Vector Institute
{rfang32,jzhao786,rpu2,jcai336,zli3446,charles.ling}@uwo.ca
runner21st@gmail.com  libinghe@msu.edu  bwang@csd.uwo.ca

## Abstract

Graph domain adaptation (GDA) transfers knowledge from a labeled source graph to an unlabeled target graph to alleviate label scarcity. In multi-view graphs, the challenge of mitigating domain shift is constrained by structural information across various views. Moreover, within each view, structures at different hops capture distinct neighborhood levels, which can lead to varying structural discrepancies. However, existing methods typically assume only a single-view graph structure, which cannot effectively capture the rich structural information in multi-relational graphs and hampers adaptation performances. In this paper, we tackle the challenging Multi-view Graph Domain Adaptation (MGDA) problem by proposing Structural Aggregation Guided Alignment (SAGA) that aligns multi-view graph data via dynamic view and neighborhood order selection. Specifically, we propose the notion of Structural Aggregation Distance (SAD) as a dynamic discrepancy metric that jointly considers view and neighborhood order, allowing the dominant view–order pair to vary during training. Through empirical analysis, we justify the validity of SAD and show that domain discrepancy in MGDA is largely governed by the dominant view–order pair, which evolves throughout training. Motivated by this observation, we design SAGA, which leverages SAD to dynamically identify the principal view-order pair that guides alignment, thereby effectively characterizing and mitigating both view- and hop-level structural discrepancies between multi-view graphs. Experimental results on various multi-relational graph benchmarks verify the effectiveness of our method. The source code is available at https://github.com/f1shungry/SAGA.

## 1 Introduction

Multi-view graphs, which involve a set of nodes with multiple relations, are prevalent in the real world because of their extraordinary ability in characterizing complex systems (Qu et al., 2017). Some typical instances include citation networks, social networks, and knowledge graphs (Wu et al., 2023b; Pan & Kang, 2023), which have multi-relational structures (Xia et al., 2025; Yu et al., 2024; 2025c; Yang et al., 2023b). Recently, this challenge has been particularly acute in multi-view graphs, where annotating structured data is especially difficult and costly, leading to pervasive label scarcity (Xu et al., 2022b; Huang et al., 2025a; Zhuo et al., 2025). To address this challenge, Multi-view Graph Domain Adaptation (MGDA) has emerged as an effective paradigm to transfer knowledge from labeled multi-view source graphs to unlabeled multi-view target graphs (Chen et al., 2019; Shi et al., 2024).

---

*Corresponding author

As shown in Fig. 2, unlike other single-view graphs, the differences in multi-view graphs stem from their diverse topological structures. Therefore, during MGDA, it is essential to consider both domain shift and cross-view shift. Early works overlook capturing multiple relation structures, a widely existing aspect of graph data. Furthermore, existing techniques fail to capture structural dependencies across multiple graph hops (Yang et al., 2024a; 2023c; 2024b; Huang et al., 2023; 2024a), thereby limiting their ability to estimate structural distributional discrepancies. Thus, a natural question arises: ($\mathcal{Q}$) **How can we quantify structural disparities across different hops between multi-view source and target graphs?** To address $\mathcal{Q}$, we introduce the Structural Aggregation

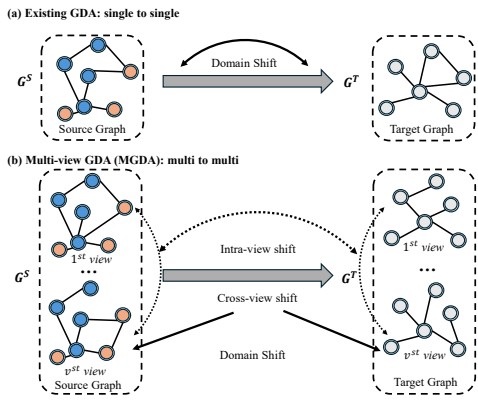

Figure 1: Illustration of the graph domain adaptation (GDA) task and MGDA (Mutli-view GDA).

Distance (SAD), a simple yet effective metric for evaluating structural disparities across various hops. Unlike prior approaches that measure discrepancies only at the first hop level (Fang et al., 2025b), SAD explicitly incorporates both the aggregation process and the distribution of node attributes to capture hop-specific differences. Empirical studies conducted on real-world datasets justify the efficacy of this novel metric in evaluating the quality of views in multi-view graphs. As shown in Fig. 3, a smaller Structural Aggregation Distance (SAD) generally indicates small cross-network performance gap between domains. Moreover, since the optimal performance may occur at different hops, each associated with the minimum SAD discrepancy, an effective approach requires dynamically aligning structural information across both view and hops during training.

To address the aforementioned challenge, we propose Structural Aggregation Guided Alignment (SAGA), which leverages the dominant view identified as the one achieving the minimum SAD across hops to guide the alignment process. Specifically, SAGA dynamically explores unsupervised view alignment during training, adaptively adjusting views and hop-wise aggregation so as to align structural information across different views. This dynamic hop alignment ensures that representations are not constrained to a fixed hop level or view, but instead are adaptively synchronized in order to capture the view and hop-specific structural disparities. Afterward, by leveraging the embedding with corresponding source and target dominant view-hops structures to guide both cross-view and intra-view alignment, we aim to mitigate domain shift as well as view shift.

We underscore the primary contributions of this study as follows:

- We first investigate Multi-View Graph Domain Adaptation (MGDA). To this end, we propose the Structural Aggregation Distance (SAD), a novel metric that quantifies structural disparities across different hops, thereby enabling a comprehensive evaluation of structural shift.

- Our empirical observations justify the design of SAD, upon which we introduce SAGA. In this framework, the view with the minimum SAD is identified through dynamic view and hop selection, enabling dominant-view alignment to mitigate both cross-view and intra-view discrepancies.

- We validate our approach through extensive experiments on extensive real-world datasets. The proposed SAGA framework consistently achieves state-of-the-art performance, demonstrating its superiority in addressing the challenges of multi-view graph domain adaptation.

## 2 RELATED WORK

### 2.1 MULTI-VIEW GRAPH LEARNING

Unsupervised domain adaptation is a wildly used setting of transfer learning methods that aims to minimize the discrepancy between the source and target domains. To solve cross-domain classification tasks, these methods are based on deep feature representation (Zhu et al., 2022; Shen et al., 2023; Yang et al., 2023a; 2022; 2025b; Xu et al., 2023a; 2022a; Yang et al., 2025a), which maps different domains into a common feature space. Some recent studies have overcome the imbalance of domains and the label distribution shift of classes to transfer model well (Zeng et al., 2023; Jing et al., 2021;

Zeng et al., 2025; Xu et al., 2023b; Zhang & He, 2025b). Some novel settings in domain adaptation have also gotten a lot of attention, like source-free domain adaptation (SFDA) (Yang et al., 2021; Xu et al., 2025b;a), test-time domain adaptation (TTDA), and relevant robustness (Wang et al., 2022; Guo et al., 2026; Yang et al., 2025c; Min et al., 2025; Huang et al., 2024b; Xu et al., 2023a; Li et al., 2025b; Ji et al., 2025; 2024). As for graph-structured data, several studies have been proposed for cross-graph knowledge transfer via GDA setting methods (Shen & Chung, 2019; Dai et al., 2022; Shi et al., 2024; Shen et al., 2025). AVDNE (Shen et al., 2020a) adopt k-hop PPMI matrix to capture high-order proximity as global consistency for source information on graphs. CDNE (Shen et al., 2020b) learning cross-network embedding from source and target data to minimize the maximum mean discrepancy (MMD) directly. GraphAE (**?**) analyzes node degree distribution shift in domain discrepancy and solves it by aligning message-passing routers. DM-GNN (Shen et al., 2023; Huang et al., 2025b; Fang et al., 2022) proposes a method to propagate node label information by combining its own and neighbors' edge structure. Furthermore, some recent research work focuses on multi-view learning in federated learning and remote sensing (Zheng et al., 2025c;b;a).

## 2.2 GRAPH DOMAIN ADAPTATION

UDAGCN (Wu et al., 2020; Yang et al., 2026) develops a dual graph convolutional network by jointly capturing knowledge from local and global levels to adapt it by adversarial training. ASN (Zhang et al., 2021) separates domain-specific and domain-invariant variables by designing a private en-coder and uses the domain-specific features in the network to extract the domain-invariant shared features across networks. SOGA (Mao et al., 2024) first time uses discriminability by encouraging the structural consistencies between target nodes in the same class for the SFDA in the graph. GraphAE (Guo et al., 2022) focuses on how shifts in node degree distribution affect node embeddings by minimizing the discrepancy between router embedding to eliminate structural shifts. SpecReg (You et al., 2022) used the optimal transport-based GDA bound for graph data and discovered that revising the GNNs' Lipschitz constant can be achieved by spectral smoothness and maximum frequency response. ALEX (Yuan et al., 2023) first creates a label shift enhanced augmented graph view using a low-rank adjacency matrix obtained through singular value decomposition by driving contrasting loss. SGDA (Qiao et al., 2023) enhances original source graphs by integrating trainable perturbations (adaptive shift parameters) into embeddings by conducting adversarial learning to simultaneously train both the graph encoder and perturbations, to minimize marginal shifts. Identifying the differences between the target and source graphs in GDA is crucial. For graph-structured data, several studies have explored cross-graph knowledge transfer using graph domain adaptation (GDA) methods (Shen & Chung, 2019; Dai et al., 2022; Shi et al., 2024). Some graph information alignment-based methods (Shen et al., 2020a;b; Yan & Wang, 2020; Shen et al., 2023; Fang et al., 2026; Xie et al., 2025a) adapt graph source node label information by integrating global and local structures from both nodes and their neighbors. UDAGCN (Wu et al., 2020) introduces a dual graph convolutional network that captures both local and global knowledge, adapting it through adversarial training. Furthermore, ASN and GraphAE (Zhang et al., 2021; Guo et al., 2022) consider extracting and aligning graph-specific information like node degree and edge shift, enabling the extraction of shared features across networks. SOGA (Mao et al., 2024) is the first to incorporate discriminability by promoting structural consistency between target nodes of the same class, specifically for source-free domain adaptation (SFDA) on graphs. SpecReg (You et al., 2022) applies an optimal transport-based GDA bound and demonstrates that revising the Lipschitz constant of GNNs can enhance performance through spectral smoothness and maximum frequency response. JHGDA (Shi et al., 2023) tackles hierarchical graph structure shifts by aggregating domain discrepancies across all hierarchy levels to provide a comprehensive discrepancy measure. ALEX Yuan et al. (2023) creates a label-shift-enhanced augmented graph view using a low-rank adjacency matrix obtained through singular value decomposition, guided by a contrasting loss function. SGDA (Qiao et al., 2023) incorporates trainable perturbations (adaptive shift parameters) into embeddings via adversarial learning, enhancing source graphs and minimizing marginal shifts. PA (Liu et al., 2024) mitigates structural and label shifts by recalibrating edge weights to adjust the influence among neighboring nodes, addressing conditional structure shifts effectively. And other graph foundation models (Wang et al., 2025b;a; Shen et al., 2024b; Xie et al., 2025b; Li et al., 2025a) focus on solving multi-domain problems.

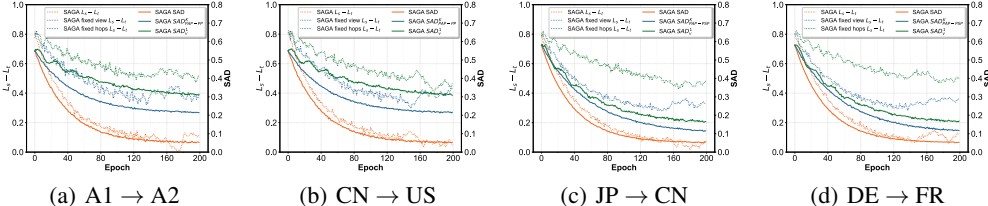

(a) A1 → A2    (b) CN → US    (c) JP → CN    (d) DE → FR

Figure 3: We report SAD performance in ACM and MAG datasets. Training dynamics of SAGA under different settings. The $x$-axis denotes training epochs, the left $y$-axis represents the classification loss discrepancy between source and target domains $(L_s - \hat{L}_t)^1$, and the right $y$-axis shows the Structural Aggregation Distance (SAD). Only the dynamic SAD consistently decreases while simultaneously driving down the source–target loss gap, indicating effective alignment. In contrast, fixing hops or views yields suboptimal results, where SAD may decrease but the loss discrepancy does not maintain a sustained reduction.

## 3 EMPIRICAL STUDY

**Definition 1 (*Mutliview Graph Domain Adaptation*).** We consider an undirected multi-relational graph $G = \{A^v, \mathcal{V}, \mathcal{E}_1, \cdots, \mathcal{E}_V, X, Y\}$, where $\mathcal{V}$ denotes the node set with $N$ nodes, and $\mathcal{E}_v$ represents the edge set in the $v$-th view. $X \in \mathbb{R}^{N \times d_f}$ is the feature matrix, and $Y$ is the label matrix with $C$ classes. For each view $v$, $A^v$ is the adjacency matrix, $D^v$ the degree matrix, and the normalized adjacency with self-loops is $\hat{A}^v = (D^v + I)^{-\frac{1}{2}}(A^v + I)(D^v + I)^{-\frac{1}{2}}$. We denote the source and target graphs as $G^S = \{A^{S,1}, \cdots, A^{S,v_S}, X^S, Y^S\}$ and $G^T = \{A^{T,1}, \cdots, A^{T,v_T}, X^T\}$, respectively. Labels (i.e., $Y_S$ and $Y_T$) express the same meanings. $Y_T$ **is fully unknown, which indicates an unsupervised problem in the target mutli-view graph.** For convenience, we define $v_T \in V_T$, $v_S \in V_S$, $k_T \in K_T$ and $k_S \in K_S$.

**Structural Aggregation Distance (SAD).** To quantify view discrepancies across different hops in domain shifts, we define the Structural Aggregation Distance(SAD). Following the aggregation rule of SGC Wu et al. (2019), the $K$-hop aggregated features of the $v$-th view are computed as:

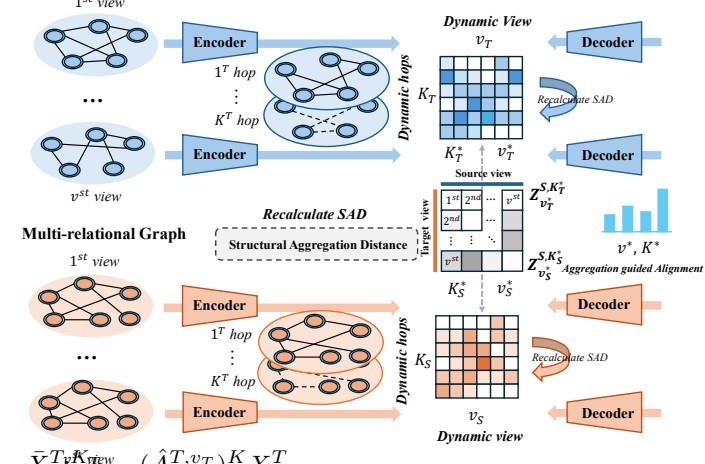

$$\bar{X}_{v_S}^{S,K_S} = (\hat{A}^{S,v_S})^K X^S, \qquad \bar{X}_{v_T}^{T,K_T} = (\hat{A}^{T,v_T})^K X^T. \tag{1}$$

Let $\bar{X}_s^{v,K}$ and $\bar{X}_t^{v,K}$ denote the aggregated centroids of source and target nodes' features after structural aggregation. Then, the $K$-hop SAD is defined as

Figure 2: Overview of the SAGA framework. It encodes multi-view graphs, dynamically selects the dominant view–hop pair via SAD, and aligns source–target embeddings under unified optimization.

$$\mathrm{SAD}_v^K = \Big\| \frac{1}{N_t} \sum_{\bar{x}_i \in \bar{X}_{v_S}^{S,K}} \bar{x}_i - \frac{1}{N_s} \sum_{\bar{x}_j \in \bar{X}_{v_T}^{T,K}} \bar{x}_j \Big\|. \tag{2}$$

Intuitively, $\mathrm{SAD}_v^K$ measures the structural discrepancy between source and target distributions in view $v_T$ and $v_S$ at hop $K_T \in \{1, \cdots, 5\}, K_S \in \{1, \cdots, 5\}$. However, given the above definition, SAD cannot be directly applied to assess view quality during training. To address this limitation, we

---

[1]Details Difinition of $\hat{L}_t$ is on in Section G of Appendix.

propose a variant method that quantifies structural discrepancy in the embedding space throughout the training process.

$$\text{SAD}_v^k = \|Z_{v_S}^{T,k_S}(Z_{v_S}^{T,k_S})^\top - Z_{v_T}^{S,k_T}(Z_{v_T}^{S,k_T})^\top\|_F^2, \tag{3}$$

where $Z_{v_T}^{T,k_T} \in Z_{V_T}^{T,K_T}$ and $Z_{v_S}^{S,k_S} \in Z_{V_S}^{S,K_S}$ are $X_{v_T}^{T,k_T} \in X_{V_T}^{T,K_T}$ and $X_{v_S}^{S,k_S} \in X_{V_S}^{S,k_S}$ embedding during training procedures and $v = \{v_S, v_T\}$, $k = \{k_S, k_T\}$. In our definition,

$$\text{SAD} = \min_{v,K} \text{SAD}_v^k. \tag{4}$$

The experimental setup follows the main experiment, with SAD adjusted into two variants: $\text{SAD}_{PSP \to PAP}^K$, which fixes $v$ as $PAP \to PSP$ in ACM dataset and $PAP \to PP$ in MAG dataset, and $\text{SAD}_V^1$, which fixes $K = 1$ with dynamic views. In Fig. 3 setting, we fix $v$ as $PSP - PAP$ and $K$ as 1 [2].

Based on the Fig. 3, we observe that dynamically adjusting SAD yields the best performance, as it most effectively reduces the discrepancy between the source and target domains. In contrast, fixing either the hops or the views for alignment results in suboptimal outcomes. Even when the discrepancies within fixed hops or views continue to decrease, the overall shift between the source and target domains does not consistently decrease.

## 4 METHOLOGY

Building on the empirical evidence in Section 3, we now present our **Structural Aggregation Guided Alignment (SAGA)** framework. Our methodology follows a progressive design: (i) obtain scalable graph embedding for downstream operation, (ii) leverage SAD to evaluate structural disparities across views and hops to identify the dominant view as the guide for adaptation,

### 4.1 SCALABLE GRAPH ENCODING

Unlike most GNN-based approaches (Mo et al., 2023a;b), we decouple graph propagation and dimensionality reduction to improve scalability. Initially, we perform propagation on the node features separately for each view, acquiring view-specific aggregated features. Similar to the approximate personalized propagation in Chien et al. (2020), we introduce the features of the original node as a teleport vector in each layer of the propagation process:

$$X^{v,0} = X, \quad X^{v,k+1} = (1-\alpha)\hat{A}^v X^{v,k} + \alpha X \tag{5}$$

where $X$ acts as both the starting matrix and the teleport set for each view. The hyper-parameter $\alpha \in [0, 1]$ represents the teleport probability. $k \in [0, K-1]$ and the aggregated features $X^v = X^{v,K}$. These features are then fed into a shared encoder for dimensionality reduction:

$$Z^v = f_\Theta(X^v) \tag{6}$$

where $Z^v \in \mathbb{R}^{N \times d_r}$ denotes node representations in the $v$-th view. This setup avoids the time-consuming graph convolution operations during training. Subsequently, the representations for each view are fed into a shared decoder for the reconstruction of view-specific aggregated features. Effective training of each view is ensured by optimizing the reconstruction MSE error:

$$\hat{X}^v = g_\Theta(Z^v) \tag{7}$$

$$\mathcal{L}_R = \frac{1}{NV} \sum_{v=1}^{V} \sum_{i=1}^{N} \|X_i^v - \hat{X}_i^v\|^2 \tag{8}$$

where the encoder $f_\Theta(\cdot)$ and the decoder $g_\Theta(\cdot)$ are both implemented using a Multilayer Perceptron (MLP) in this study. After applying the scalable graph encoding to extract graph information, the output node embedding for the source and target graph is denoted as $Z_{V_S}^{S,K_S} = \{Z_1^{S,1}, \cdots, Z_{v_S}^{S,k_s}\}$, $Z_{V_T}^{T,K_T} = \{Z_1^{T,1}, \cdots, Z_{v_T}^{T,k_T}\}$.

---

[2]Details analysis on the $K$ are presented in Section F of Appendix.

## 4.2 DYNAMIC NEIGHBORHOOD ORDER SELECTION VIA SAD

For each view $v$ and hop $K \in \{1, \ldots, K_{\max}\}$, we measure the structural discrepancy as $SAD$. In practice, we usually set $K_{\max} = 5$. The dominant pair $(v_S^*, K_S^*), (v_T^*, K_T^*)$ is defined as

$$v^*, k^* = \arg\min_{v,K} \text{SAD}_v^k, \tag{9}$$

where $v^* = (v_S^*, v_T^*)$, $k^* = (k_T^*, k_T^*)$ provides the most transferable structural signal during training.

**SAD guided weight.** Rather than placing weights directly on the hop index $K$, we assign weights to the corresponding discrepancies $\text{SAD}_v^k$, such that smaller values receive higher emphasis. Formally, we define

$$\omega_{v,k} = \frac{\exp(-\lambda \, \text{SAD}_v^k)}{\sum_{j=1}^{K} \sum_{i=1}^{V} \exp(-\lambda \, \text{SAD}_i^j)}, \qquad \lambda > 0, \tag{10}$$

where $\omega_{v,k} = \{\omega_{v_S, k_S}, \omega_{v_T, k_T}\}$ and the temperature parameter $\lambda$ controls the sharpness of the weighting. This construction ensures that the hop with the minimal $\text{SAD}_v^k$ naturally obtains the largest weight, while still allowing other hops to contribute proportionally to their discrepancies.

**Intra-alignment to the dominant view.** For cross-view alignment, we employ the discrete optimal hop $k_v^* = \arg\min_K \text{SAD}_v^K$ for each view and the global dominant pair $(v^*, k^*)$, and minimize

$$\mathcal{L}_{IA} = \frac{1}{N_T^2 V_T} \sum_{v=1}^{V_T} \omega_{v_T, k_T} \left\| Z_{v_T}^{k^*} \left(Z_{v_T}^{k_T^*}\right)^\top - Z_{v_T}^{T,1} \left(Z_{v_T}^{T,1}\right)^\top \right\|_F^2 + \frac{1}{N_S^2 V_S} \sum_{v=1}^{V_S} \omega_{v_S, k_S} \left\| Z_{v_S}^{k_S^*} \left(Z_{v_S}^{k_S^*}\right)^\top - Z_{v_S}^{S,1} \left(Z_{v_S}^{S,1}\right)^\top \right\|_F^2, \tag{11}$$

where $Z_{v_T}^T$ and $Z_{v_S}^S$ denote the embedding matrices for view $v_T$ in the target domain and view $v_S$ in the source domain, respectively (computed after $k$ aggregation hops); to reduce computational cost, we set $k = 1$ by default in training stage. Only calculate other hops in obtain SAD. During $\mathcal{L}_{IA}$, it can be constrained through aligning the each dominant view space. To this end, it can obtain a uniform dominate guided embedding.

**Cross-alignment to the dominant view.**

$$\ell\left(Z_{v_S^*}^{S,k_S^*}, Z_{v_T^*}^{T,k_T^*}\right) = -\log \frac{1}{N_T N_S} \sum_{j=1}^{N_T} \sum_{i=1}^{N_S} \exp\left(\cos\left(\tilde{z}_{i,v_S^*}^{S,k_S^*}, \tilde{z}_{j,v_T^*}^{T,k_T^*}\right)/\tau\right) \tag{12}$$

where $\tilde{Z}_{v_S^*}^{S,k_S^*}$ and $\tilde{Z}_{v_T^*}^{T,k_T^*}$ is the non-linear projection of $Z_{v_S^*}^{S,k_S^*}$ and $Z_{v_T^*}^{T,k_T^*}$. $\cdot \text{sim}(\cdot)$ refers to the cosine similarity and $\tau$ is the temperature parameter. To integrate the dominant view and dynamically aligned hops, we adopt a contrastive learning objective. Representations from each view are projected into a shared latent space, and aligned with the dominant pair view:

$$\mathcal{L}_{CA} = \ell(Z_{v_S^*}^{S,k_S^*}, Z_{v_T^*}^{T,k_T^*}) + \ell(Z_{v_T^*}^{T,k_T^*}, Z_{v_S^*}^{S,k_S^*}), \tag{13}$$

where $\ell(\cdot, \cdot)$ is the alignment loss based on cosine similarity. By applying these loss, we align both intra-domain and cross-domain embeddings.

## 4.3 TARGET NODE CLASSIFICATION

The source classifier loss $\mathcal{L}_S\left(f_S\left(Z^S\right), Y^S\right)$ is to minimize the cross-entropy for the labeled data node in the source domain:

$$\mathcal{L}_S\left(f_S\left(Z^S\right), Y^S\right) = -\frac{1}{N_S} \sum_{i=1}^{N_S} y_i^S \log\left(\hat{y}_i^S\right) \tag{14}$$

where $y_i^S$ denotes the label of the $i$-th node in the source domain and $\hat{y}_i^S$ are the classification prediction for the $i$-th source graph labeled node $v_i^S \in \mathcal{V}^S$. To utilize the data in the target domain, we use entropy loss for the target classifier $f_T$:

$$\mathcal{L}_T\left(f_T\left(Z^T\right)\right) = -\frac{1}{N_T} \sum_{i=1}^{N_T} \hat{y}_i^T \log\left(\hat{y}_i^T\right) \tag{15}$$

| Methods | Metric | CN→US | US→CN | JP→CN | CN→JP | DE→FR | FR→DE | RU→US | US→RU | DE→CN | CN→DE |
|---|---|---|---|---|---|---|---|---|---|---|---|
| GCN | ACC | 0.351 | 0.263 | 0.327 | 0.335 | 0.319 | 0.302 | 0.301 | 0.317 | 0.281 | 0.291 |
|  | macro-F1 | 0.217 | 0.218 | 0.221 | 0.201 | 0.203 | 0.167 | 0.233 | 0.320 | 0.167 | 0.223 |
| GAT | ACC | 0.325 | 0.316 | 0.361 | 0.309 | 0.324 | 0.253 | 0.265 | 0.339 | 0.281 | 0.316 |
|  | macro-F1 | 0.218 | 0.320 | 0.253 | 0.237 | 0.212 | 0.249 | 0.221 | 0.323 | 0.249 | 0.209 |
| AdaGCN | ACC | 0.418 | 0.483 | 0.389 | 0.425 | 0.431 | 0.314 | 0.370 | 0.435 | 0.359 | 0.363 |
|  | macro-F1 | 0.231 | 0.362 | 0.257 | 0.244 | 0.242 | 0.285 | 0.242 | 0.369 | 0.265 | 0.212 |
| UDAGCN | ACC | 0.425 | 0.453 | 0.451 | 0.414 | 0.352 | 0.364 | 0.359 | 0.421 | 0.356 | 0.369 |
|  | macro-F1 | 0.243 | 0.341 | 0.271 | 0.275 | 0.274 | 0.262 | 0.239 | 0.361 | 0.296 | 0.253 |
| GRADE-N | ACC | 0.421 | 0.467 | 0.376 | 0.404 | 0.455 | 0.375 | 0.353 | 0.429 | 0.358 | 0.362 |
|  | macro-F1 | 0.241 | 0.352 | 0.305 | 0.292 | 0.286 | 0.287 | 0.245 | 0.347 | 0.299 | 0.252 |
| ACDNE | ACC | 0.410 | 0.458 | 0.422 | 0.436 | 0.463 | 0.359 | 0.351 | 0.434 | 0.360 | 0.378 |
|  | macro-F1 | 0.225 | 0.352 | 0.335 | 0.275 | 0.279 | 0.292 | 0.231 | 0.388 | 0.265 | 0.291 |
| ASN | ACC | 0.416 | 0.431 | 0.335 | 0.432 | 0.433 | 0.368 | 0.289 | 0.435 | 0.334 | 0.372 |
|  | macro-F1 | 0.232 | 0.387 | 0.219 | 0.251 | 0.248 | 0.297 | 0.216 | 0.226 | 0.258 | 0.270 |
| SpecReg | ACC | 0.425 | 0.501 | 0.363 | 0.419 | 0.424 | 0.387 | 0.328 | 0.429 | 0.330 | 0.371 |
|  | macro-F1 | 0.247 | 0.380 | 0.273 | 0.263 | 0.236 | 0.282 | 0.213 | 0.371 | 0.279 | 0.303 |
| PA | ACC | 0.402 | 0.529 | 0.351 | 0.323 | 0.416 | 0.357 | 0.306 | 0.317 | 0.336 | 0.357 |
|  | macro-F1 | 0.236 | 0.331 | 0.295 | 0.243 | 0.229 | 0.299 | 0.235 | 0.367 | 0.286 | 0.263 |
| GraphATA | ACC | 0.407 | 0.518 | 0.382 | 0.341 | 0.327 | 0.414 | 0.351 | 0.341 | 0.341 | 0.321 |
|  | macro-F1 | 0.241 | 0.339 | 0.287 | 0.252 | 0.291 | 0.289 | 0.234 | 0.351 | 0.274 | 0.254 |
| HGDA | ACC | 0.417 | 0.529 | 0.392 | 0.343 | 0.391 | 0.421 | 0.361 | 0.337 | 0.361 | 0.317 |
|  | macro-F1 | 0.239 | 0.333 | 0.321 | 0.251 | 0.284 | 0.279 | 0.212 | 0.344 | 0.271 | 0.249 |
| **SAGA** | ACC | **0.553** | **0.535** | **0.512** | **0.557** | **0.529** | **0.515** | **0.377** | **0.441** | **0.425** | **0.381** |
|  | macro-F1 | **0.341** | **0.426** | **0.372** | **0.415** | **0.426** | **0.401** | **0.378** | **0.397** | **0.320** | **0.350** |

Table 1: Cross-network node classification on MAG datasets

where $\hat{y}_i^T$ are the classification prediction for the $i$-th node in the target graph $v_i^T$. Finally, by combining $\mathcal{L}_A$, $\mathcal{L}_S$, $\mathcal{L}_D$ and $\mathcal{L}_T$, the overall loss function of our model can be represented as:

$$\mathcal{L} = \mathcal{L}_R + \alpha\mathcal{L}_{IA} + (1-\alpha)\mathcal{L}_{CA} + \beta\mathcal{L}_S + \delta\mathcal{L}_T \tag{16}$$

where $\alpha$, $\beta$ and $\delta$ are trade-off hyper-parameters. The parameters of the whole framework are updated via backpropagation.

# 5 EXPERIMENT

## 5.1 DATASETS

We use real-world citation network datasets for our experiments to demonstrate the effectiveness of our proposed algorithm for multi-view graph-domain adaptation in the node classification task. Specifically, we evaluate on ACM1 (Shen et al., 2024a), ACM2 (Shen et al., 2024a), and MAG (Shen et al., 2024a) to test how well the method transfers across views and domains while preserving node-class information.

In these graphs, nodes correspond to papers and edges denote citation relationships between papers. The edges are defined by two meta-paths: PAP (paper–author–paper) and PSP (paper–subject–paper) for ACM1 and ACM2, and PAP (paper–author–paper) and PP (paper–paper) for MAG.

The ACM1 and ACM2 graphs are labeled into three classes: database, wireless communication, and data mining. As the subset of OGBN-MAG (Wang et al., 2020), MAG is partitioned into six subgraphs(CN, US, DE, FR, JP, and RU), each corresponding to a country. From these country subgraphs, we extract the ten most frequent classes. Any two of these country subgraphs can be used as the source and target datasets in the multi-view graph node classification experiment. Details of the experimental datasets' statistics are provided in Table Appendix B.

## 5.2 BASELINES

In this work, we compare a representative set of baselines for multi-view graph domain adaptation (GDA) in node classification. We begin with non-adaptive methods, where **GCN** (Kipf & Welling, 2016) provides a strong single-view baseline by performing standard graph convolution without any domain adaptation, and **GAT** (Veličković et al., 2017) offers an attention-based non-adaptive alternative that leverages expressive neighbor aggregation. **AdaGCN** (Dai et al., 2022) follows as an adaptive propagation method that adjusts message passing to node-level or edge-level cues,

informing how learning dynamics respond to varying neighborhood structures in a multi-view setting. **UDAGCN** (Wu et al., 2020) represents a dual-graph convolutional-network component learning framework for unsupervised GDA, capturing knowledge from local and global levels and using adversarial training to align representations across views. **GRADE-N** (Wu et al., 2023a) introduces a graph subtree discrepancy to quantify distribution shifts between source and target graphs, enabling principled evaluation of domain-aware transfer in multi-view contexts.

**ACDNE** (Shen et al., 2020a) is an adaptation-oriented method designed to align and fuse representations across views through cross-view objectives, contributing to robust cross-view transfer, while **ASN** (Zhang et al., 2021) leverages domain-specific features within each network to extract domain-invariant shared representations across multiple networks. **SpecReg** (You et al., 2022) realizes performance improvements via regularization inspired by the intersection of optimal transport domain adaptation and graph-filter theory, thereby aiding cross-view generalization under distribution shifts. Finally, **PA** (Liu et al., 2024) mitigates distribution shifts in graph data by recalibrating edge influences to address structural shifts and by adjusting classification losses to tackle label shifts, enhancing cross-view robustness. Aggregate

| Methods | ACM1→ACM2 | | ACM2→ACM1 | |
|---|---|---|---|---|
| | ACC | F1 | ACC | F1 |
| GCN | 0.367 | 0.253 | 0.359 | 0.236 |
| GAT | 0.373 | 0.319 | 0.363 | 0.241 |
| AdaGCN | 0.431 | 0.272 | 0.386 | 0.362 |
| UDAGCN | 0.452 | 0.283 | 0.409 | 0.347 |
| GRADE-N | 0.469 | 0.291 | 0.373 | 0.343 |
| ACDNE | 0.457 | 0.261 | 0.392 | 0.363 |
| ASN | 0.436 | 0.321 | 0.437 | 0.327 |
| SpecReg | 0.493 | 0.308 | 0.431 | 0.381 |
| PA | 0.506 | 0.333 | 0.452 | 0.377 |
| GraphATA | 0.501 | 0.322 | 0.401 | 0.395 |
| HGDA | 0.511 | 0.311 | 0.441 | 0.414 |
| **SAGA** | **0.523** | **0.351** | **0.454** | **0.444** |

Table 2: Experiment on ACM datasets.

to Adapt (**GraphATA**) (Zhang & He, 2025a) performs node-wise, personalized aggregation of source GNNs for multi-source-free unsupervised graph domain adaptation, achieving state-of-the-art performance across benchmarks. **HGDA** (Fang et al., 2025b) is a framework that mitigates cross-domain discrepancies by explicitly aligning homophilic, heterophilic, and attribute signals through specialized filters, thereby improving node classification across source and target graphs.

## 5.3 EXPERIMENTAL SETUP

Our multi-view graph setup comprises multiple views for each domain (source/target). To ensure fair comparison, all baselines are evaluated under a consistent training/evaluation protocol, with explicit handling of single-view baselines in a multi-view context as described below.

In multi-view settings, baseline handling is as follows. For non-adaptation baselines (e.g., GCN (Kipf & Welling, 2016), GAT (Veličković et al., 2017)), each view in the source domain is processed independently according to its original formulation to obtain per-view embeddings; we then average the embeddings across all source views and train a GNN using these averaged embeddings with the available labels. For the target domain, we compute each view's embedding independently using the method described in the respective paper, average them, and input this averaged embedding to the trained GNN for node classification. This approach preserves the core idea of the original non-adaptive methods while leveraging cross-view aggregation for transfer. For adaptation baselines (e.g., AdaGCN (Dai et al., 2022), UDAGCN (Wu et al., 2020), GRADE-N (Wu et al., 2023a), ACDNE (Shen et al., 2020a), ASN (Zhang et al., 2021), SpecReg (You et al., 2022), PA (Liu et al., 2024)), we compute view-specific embeddings for each source view, then average the embeddings across all source views to obtain a single representative source embedding for downstream inference on the target domain. In the target domain, target-view embeddings are computed according to the method's formulation and, if required by the baseline, are aligned or regularized via the adaptation mechanism during training.Accuracy and macro-F1 score are used as evaluation metrics, as they are standard measures for classification problems.

## 5.4 RESULTS AND ANALYSES

The results of experiments are summarized in Table 1 and Table 2, where the best performance is highlighted in bold. A clear observation from the results is that our proposed SAGA method achieves the highest accuracy and macro-F1 scores across all listed source-target domain pairs, demonstrating its superior performance for multi-view graph domain adaptation in node classification compared

| Methods | ACM1→ACM2 | | ACM2→ACM1 | | Methods | CN→US | | US→CN | |
|---|---|---|---|---|---|---|---|---|---|
| | ACC | F1 | ACC | F1 | | ACC | F1 | ACC | F1 |
| $SAGA_{PSP→PAP}$ | 0.366 | 0.314 | 0.358 | 0.296 | $SAGA_{PP→PAP}$ | 0.316 | 0.183 | 0.318 | 0.276 |
| $SAGA_{PAP→PAP}$ | 0.379 | 0.319 | 0.363 | 0.283 | $SAGA_{PAP→PAP}$ | 0.329 | 0.191 | 0.346 | 0.318 |
| $SAGA_{PSP→PSP}$ | 0.351 | 0.339 | 0.373 | 0.291 | $SAGA_{PP→PP}$ | 0.335 | 0.177 | 0.323 | 0.297 |
| $SAGA_{PAP→PSP}$ | 0.351 | 0.332 | 0.389 | 0.302 | $SAGA_{PAP→PP}$ | 0.321 | 0.230 | 0.349 | 0.321 |
| **SAGA** | **0.523** | **0.351** | **0.454** | **0.443** | **SAGA** | **0.553** | **0.341** | **0.535** | **0.426** |

Table 3: View Ablation studies on ACM and MAG datasets.

to all baseline methods.The unsatisfactory performance of GCN (Kipf & Welling, 2016)and GAT (Veličković et al., 2017) in this task indicates that merely transferring knowledge learned from the source domain to the target domain is inadequate for domain adaptation, underscoring the crucial importance of achieving proper alignment between source and target domains. While other adaptation methods demonstrate improvements over non-adaptive methods, they still fall significantly short of SAGA, suggesting that effectively analyzing and leveraging the relationships and information embedded within multiple views of multi-view graphs is essential for superior performance. In addition, we present analyses of model efficiency and hyperparameter sensitivity in Appendix C. Specifically, although our approach requires computing $N_T \times N_T$ pairwise distances, these terms do not participate in backpropagation, thereby avoiding the need to propagate gradients through all distance computations. Consequently, while the method introduces some computational overhead, the overall complexity remains practically acceptable. We further conduct sensitivity studies on the hyperparameters $\alpha$, $\beta$, $\delta$ in Appendix F, and $K$, showing that SAGA maintains stable performance under varying configurations, confirming the robustness of the model.

## 5.5 View Ablation Study

We evaluate the contribution of each relational view by restricting **SAGA** to single-view training, while keeping all other settings unchanged. The subscripts denote the source→target views used by each SAGA variant; e.g., $SAGA_{PSP→PAP}$ aligns representations from PSP to PAP. On ACM1/ACM2 we consider PAP (paper–author–paper) and PSP (paper–subject–paper); on MAG we consider PAP and PP (paper–paper). As reported in Table 3, every restricted variant yields noticeably lower ACC/F1 than the full SAGA, which aggregates and co-aligns all available views. The gaps are sizable (e.g., on ACM1→ACM2, SAGA improves ACC by 0.21 and F1 by 0.10 over the best single/cross-view setting). These results indicate that each view encodes complementary semantics that PAP captures author/co-authorship proximity, PSP captures topical similarity, and PP captures paper-level linkage and leveraging them jointly is necessary to attain the best cross-domain transfer performance.

## 5.6 Model Ablation Study

To validate the design choices of our proposed SAGA framework, we conduct a comprehensive ablation study evaluating the contribution of each key component. All experiments are performed on the ACM1 and ACM2 datasets under the multi-view domain adaptation setting. We compare the following variants of our method:

- **SAGA w/o DVA**: Remove the Dominant View Alignment module ($\mathcal{L}_{IA} + \mathcal{L}_{CA}$). Replace alignment by the DANN method.
- **SAGA w/o Dynamic Hop**: Fix the hop number $K^v$ across all views instead of dynamically updating it.
- **SAGA w/o Dominant View**: Replace the dominant view selection with a simple averaging of all views.
- **SAGA (Full)**: Our complete framework with all components.

Fig. 4 demonstrates the essential contributions of each SAGA component. The dominant view alignment module ($\mathcal{L}_{DVA}$) is most critical, with its removal causing the largest performance drop. Dynamic hop selection significantly outperforms fixed-hop aggregation, confirming the need for

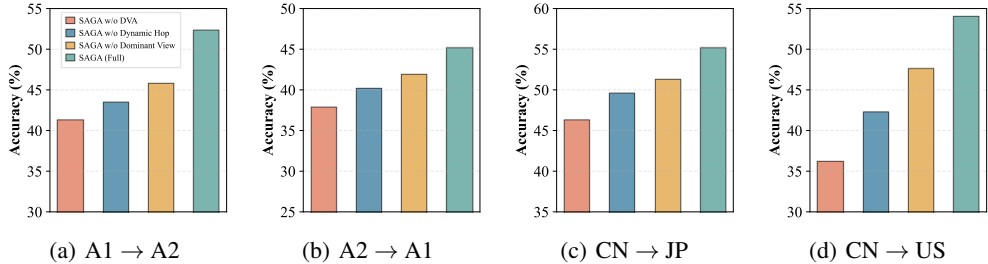

Figure 4: Classification accuracy of different variants of SAGA on ACM and MAG dataset

view-specific depth adaptation. Simply averaging views without identifying the dominant anchor also reduces performance. The full SAGA framework integrates these components, which collectively address structural discrepancies in multi-view graph domain adaptation.

# 6 CONCLUSION

In this paper, we propose the SAGA framework to address the multi-view graph domain adaptation problem in cross-domain node classification. We reveal the importance of multi-view structural alignment through SAD in empirical analysis (Zeng et al., 2024c;b;a; Pu et al., 2025). Our approach dynamically aggregates and aligns multi-view graph structures by optimizing cross-hop and cross-view structural consistency. Comprehensive experiments on multi-relational graph benchmarks verify the superiority of our approach. We will also delve deeper into multi-view graph domain adaptation theory to develop more powerful and generalizable models in many other relvent aeras. (Jing et al., 2025; Yu et al., 2025b; Jing et al., 2026b;a; Yu et al., 2025a).

# 7 ACKNOWLEDGMENTS

This work is supported by the Natural Sciences and Engineering Research Council of Canada (NSERC) Discovery Grants program.

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

| Types | Datasets | #Node | #Relation | #Edge | #Feat | #Label |
|-------|----------|-------|-----------|-------|-------|--------|
| ACM | ACM1 | 3,025 | PAP
PSP | 26,256
2,207,736 | 512 | 3 |
| | ACM2 | 4,019 | PAP
PSP | 57,853
4,338,213 | 512 | 3 |
| MAG | CN | 25,467 | PAP
PP | 157,303
43,683 | 128 | 10 |
| | US | 55,392 | PAP
PP | 12,332,742
289,833 | 128 | 10 |
| | JP | 12,651 | PAP
PP | 41,165
20,747 | 128 | 10 |
| | FR | 12,052 | PAP
PP | 51,739
30,915 | 128 | 10 |
| | DE | 17,919 | PAP
PP | 77,100
45,528 | 128 | 10 |
| | RU | 3,690 | PAP
PP | 3,315
3,539 | 128 | 10 |

Table 4: Statistics of experimental datasets

## A    USE OF LARGE LANGUAGE MODELS (LLMS)

We declare that Large Language Models (LLMs) were used exclusively as auxiliary tools to aid the writing of this paper, primarily for polishing, grammar checking, and improving readability. LLMs were not involved in conceptual design, theoretical formulation, experimental implementation, or result analysis. All research contributions, including methodology, experiments, and conclusions, were conceived, conducted, and validated entirely by the authors.

## B    DATASET EXPERIMENT DETAILS

The ACM1 and ACM2 graphs are labeled into three classes: database, wireless communication, and data mining. As the subset of OGBN-MAG  (Wang et al., 2020), MAG is partitioned into six subgraphs(CN, US, DE, FR, JP, and RU), each corresponding to a country. From these country subgraphs, we extract the ten most frequent classes. Any two of these country subgraphs can be used as the source and target datasets in the multi-view graph node classification experiment. Details of the experimental datasets' statistics are provided in Table 4.

## C    MODEL EFFICIENT EXPERIMENT

**Model Complexity**: Here, we analyze the computational complexity of the proposed SAGA. The computational complexity primarily depends on the filter layers and key loss function. For a given graph $G$, let $N$ represent the total number of nodes in the graph, $V$ represent view number and $d$ the feature dimension. For Scalable Graph Encoding $f_\Theta$: Includes matrix multiplication with $O(N^2 \cdot d)$. SAGA per-iteration complexity is dominated by multi-view propagation $O(VN \cdot d)$, and a forward-only SAD computation $O(VN^2 d)$. Combining these yields an overall complexity of $O\left(VN \cdot d + VN^2 \cdot d\right)$, where the quadratic SAD term does not incur gradient cost, making the practical training overhead similar to existing GDA methods. Thus, the total complexity of SAGA is still: $O(VN^2 \cdot d)$. Note that HGDA is inherently a single-view method. If we extend HGDA to the multi-view MGDA setting, it would inherit the same computational overhead, resulting in a complexity of $O(VN^2 \cdot d)$. Therefore, although SAGA incorporates dynamic multi-view selection, its computational overhead remains comparable to that of HGDA; the two methods do not differ by orders of magnitude in asymptotic complexity.

**Model Efficient Experiment**: To further investigate the efficiency of SAGA, Table4 reports the running time comparison across various algorithms. We also compared the training time and GPU memory usage of common baselines UDAGCN and a recent SOTA method, ASN, and HGDA. As

| Dataset | Method | Training Time (Normalized w.r.t. UDAGCN) | Memory Usage (Normalized w.r.t. UDAGCN) | Accuracy(%) |
|---------|--------|------------------------------------------|------------------------------------------|-------------|
| ACM1→ACM2 | UDAGCN | 1 | 1 | 0.452 |
| | ASN | **1.314** | **1.414** | 0.436 |
| | PA | 1.181 | 1.171 | 0.556 |
| | HGDA | 0.914 | 1.124 | 0.521 |
| | *SAGA* | 1.163 | 1.112 | **0.793** |
| CN→US | UDAGCN | 1 | 1 | 0.425 |
| | ASN | **1.413** | **1.513** | 0.402 |
| | PA | 1.011 | 1.103 | 0.402 |
| | HGDA | 1.007 | 1.173 | 0.417 |
| | *SAGA* | 1.109 | 1.098 | **0.430** |
| B→E | UDAGCN | 1 | 1 | 0.510 |
| | ASN | **1.311** | **1.173** | 0.569 |
| | PA | 1.102 | 1.097 | 0.562 |
| | HGDA | 1.051 | 1.032 | 0.566 |
| | *SAGA* | 1.148 | 1.107 | **0.573** |

Table 5: Comparison of Training Time, Memory Usage, and Accuracy on ACM and MAG datset.

shown in Table 5, the evaluation results on ACM dataset further demonstrate that our method achieves superior performance with tolerable computational and storage overhead.

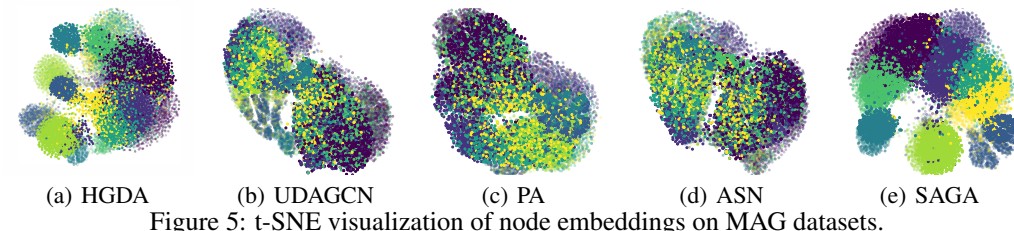

(a) HGDA     (b) UDAGCN     (c) PA     (d) ASN     (e) SAGA

Figure 5: t-SNE visualization of node embeddings on MAG datasets.

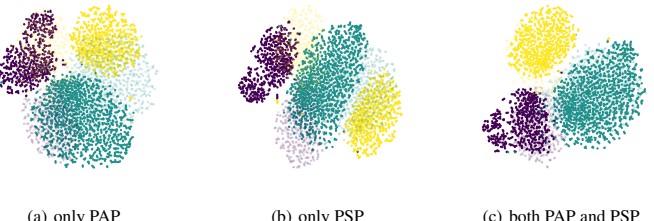

(a) only PAP     (b) only PSP     (c) both PAP and PSP

Figure 6: t-SNE visualization of node embeddings on ACM1 and ACM2 datasets. (a) Embeddings using only the PAP view; (b) Embeddings using only the PSP view; (c) Embeddings using the joint PAP and PSP views. Each color represents a distinct class, with lighter and darker shades indicating source and target domain nodes within the same class, respectively.

# D    VISUALIZATION

To evaluate the quality of learned representations, we visualize the node embeddings generated by SAGA using t-SNE on both ACM1 and ACM2 datasets. We compare three different configurations: (a) embeddings using only the PAP structural view, (b) embeddings using only the PSP structural view, and (c) embeddings incorporating both PAP and PSP views (joint representation).

As shown in Fig.6, each color represents a distinct node class, while within each class, lighter and darker shades denote nodes from the source and target domains, respectively. All three configurations achieve clear separation between different classes, demonstrating that SAGA effectively preserves inter-class discriminability.

However, significant differences emerge in the alignment of cross-domain nodes within the same class. In subfigures (a) and (b), where only single views are used, source and target domain nodes of the same class exhibit noticeable distribution gaps. In contrast, subfigure (c) shows substantially

improved overlap between domains within each class, indicating that the joint representation better aligns the source and target distributions.

These results suggest that integrating multiple structural views provides complementary information that enhances domain adaptation performance. The multi-view approach not only maintains class discrimination but also facilitates better cross-domain alignment, ultimately yielding more transferable node representations for classification tasks.

To further demonstrate the advantage of our proposed method, we also visualize the embedding results on MAG generated by competitive methods HGDA, UDAGCN, PA, and ASN, which are shown in Fig.5. Our SAGA method produces a compact alignment structure. In other words, our method has the best domain distance between source target and the most distinct boundaries between different classes.

## E   DESCRIPTION OF ALGORITHM SAGA

---

**Algorithm 1:** The proposed algorithm SAGA

---

**Input:** Source node feature matrix $X^S$; source original graph adjacency matrix $A^{S,V}$; Target node feature matrix $X^T$; Target graph adjacency matrix $A^{T,V}$ source node label matrix $Y^S$; maximum number of iterations $\eta$

Compute the $SAD_v^k$ between source and target graph by running Formula 3 algorithm.

**for** $it = 1$ **to** $\eta$ **do**

 $Z_{V_S}^{S,K_S} = Encoder(A^{S,V_S}, X^S)$

 $Z_{V_S}^{T,K_T} = Encoder(\hat{A}^S, X^S)$// embedding of source and target graph

 $X^S = Decoder(Z_{V_S}^{S,K_S})$

 $X^T = Decoder(Z_{V_S}^{T,K_T})$// decoding of source and target graph

 $L_R$to alleviate representation collapse

 $SAD = \min_{v,K} SAD_v^k$. is calculated once per epoch

 $Z_{V_S}^{S,K_S}$ and $Z_{V_S}^{T,K_T}$ through intra-view alignment by Formula 11 algorithm.

 $Z_{V_S}^{S,K_S}$ and $Z_{V_S}^{T,K_T}$ through cross-view alignment by Formula 13 algorithm.

 Domain adaptive between $S^S$ and $S^T$// adaptive in both cross and intra views

 $\hat{y}_i^S$constrained by$y_i^S$ and $\hat{y}_i^T$ constrained by$\hat{y}_i^T$

 Calculate the overall loss with Eq.(16)

 Update all parameters of the framework according to the overall loss

**end**

Predict the labels of target graph nodes based on the trained framework.

**Output:** Classification result $\hat{Y}^T$

---

## F   PARAMETER ANALYSIS

$\alpha$ is chosen from the set $\{0.05, 0.25, 0.5, 0.75, 0.95\}$ while $\beta$ and $\delta$ are chosen from the set $\{0.005, 0.01, 0.1, 0.5, 1.5\}$. These values provide flexibility for adjusting the relative importance of different loss terms. $K$ (the number of neighbors order) selection is typically $K \in \{1, \cdots, 10\}$ . The optimal value for $k$ depends on the density and connectivity of the graph. Due to extremely large $k$ could also introduce noisy that will deteriorate the performance. Usually our largest $k$ will be 5.

The ACM dataset typically exhibits fewer nodes but complex edge structures. Due to its sparsity, relatively larger values of $\alpha$, $\beta$, and $\delta$ are employed to enhance topology alignment and capture essential structural relations, with parameter $\beta$ selected from $\{0.1, 0.5\}$ and $\alpha$ selected from $\{0.5, 0.25\}$. Moreover, smaller neighborhood orders are preferred, and $k$ is chosen from $3, 4$ accordingly.

MAG Dataset: The MAG dataset is large and diverse, containing losts of classes with various relationships and rich metadata. Structural and attribute alignment are key factors. In this context, attribute shifts have a significant impact on the values of $\alpha, \beta$ and $\delta$, the majority of which are selected

| Types | Datasets | $\alpha$ | $\beta$ | $\delta$ | $k$ |
|---|---|---|---|---|---|
| ACM | ACM1→ACM2 | 0.5 | 0.5 | 0.01 | 4 |
| | ACM2→ACM1 | 0.25 | 0.1 | 0.01 | 3 |
| MAG | CN→US | 0.5 | 0.1 | 0.1 | 5 |
| | US→CN | 0.25 | 0.1 | 0.1 | 5 |
| | JP→CN | 0.25 | 0.5 | 0.1 | 4 |
| | CN→JP | 0.25 | 0.1 | 0.5 | 5 |
| | DE→FR | 0.25 | 0.1 | 0.1 | 4 |
| | FR→DE | 0.25 | 0.1 | 0.01 | 5 |
| | RU→US | 0.25 | 0.1 | 0.1 | 4 |
| | US→RU | 0.25 | 0.1 | 0.01 | 5 |
| | DE→CN | 0.5 | 0.1 | 0.1 | 5 |
| | CN→DE | 0.25 | 0.01 | 0.1 | 5 |

Table 6: Experiment hyperparameter setting Value.

from the set $\{0.1, 0.25, 0.5\}$. The parameter $k$ works well in this context, enabling the model to capture high-level local and global structural information within the graph. We select $k$ from $\{4, 5\}$.

In this section, we analyze the sensitivity of the parameters of our method on the ACM dataset and MAG dataset. As shown in Figure.7 in each subfigure, the accuracy usually peaks at $2-3$ with $k$. This is reasonable since increasing $k$ means more high-order proximity information is incorporated. On the other hand, extremely large $k$ could also introduce noise that will deteriorate the performance. Furthermore, each dataset may require a unique optimal value for $k$, determined by its inherent properties. From Figure.8 subfigures for $\alpha$, $\beta$, and $\delta$, we can see SAGA has competitive performance on a large range of values, which suggests the stability of our method.

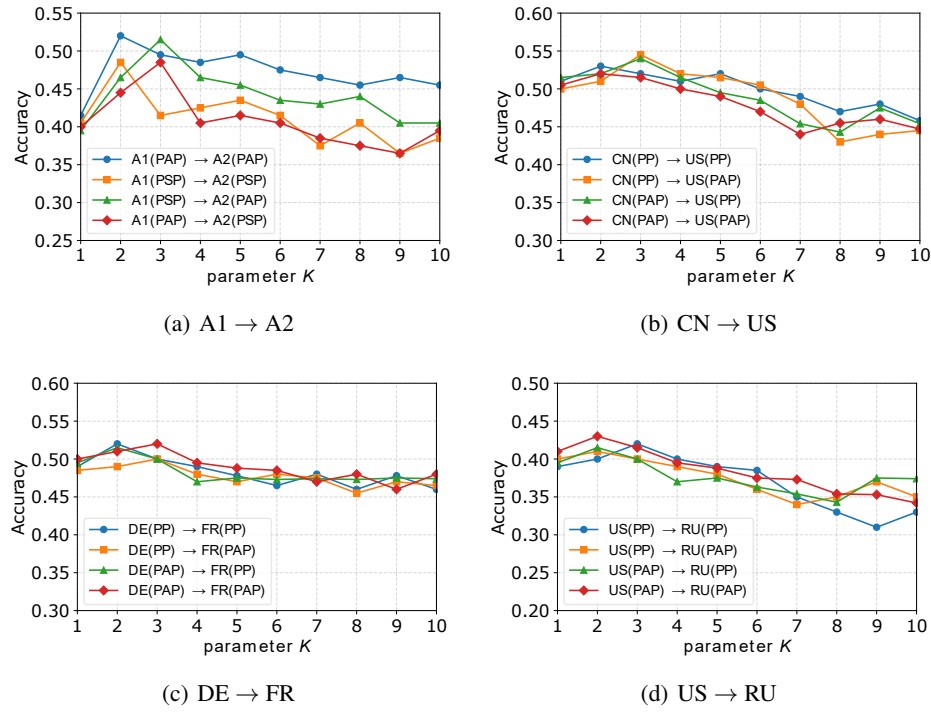

Figure 7: The influence of parameters $k$ on ACM and MAG datasets.

## G EXPERIMENT DETAIL OF EMPIRICAL STUDY

We evaluate the proposed SAGA framework under a unified multi-view graph domain adaptation protocol on both the ACM and MAG benchmarks. To quantify the performance gap between the

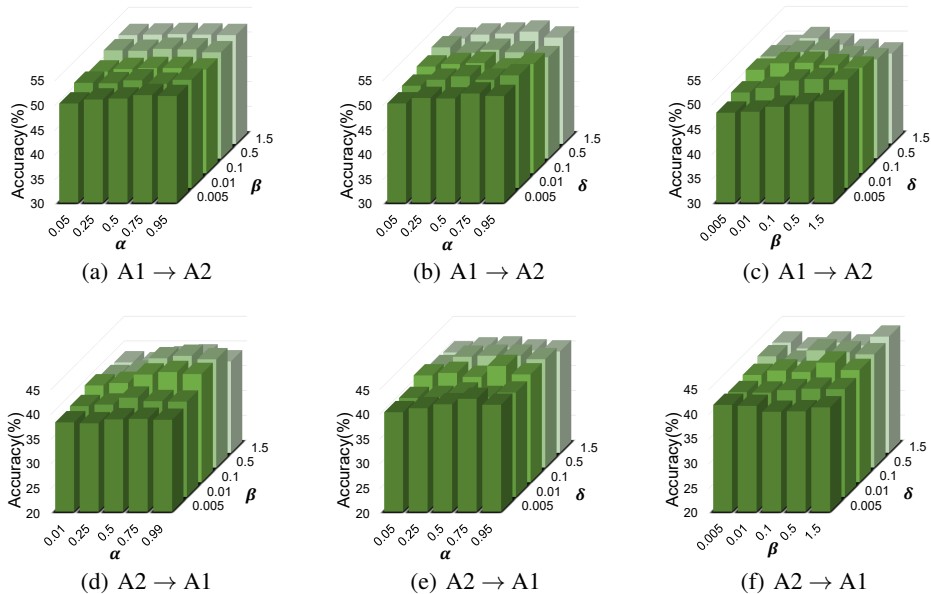

Figure 8: The influence of parameters $\alpha$, $\beta$, and $\delta$ on ACM datasets.

source and target domains, we compute $L_s - \hat{L}_t$. Specifically, $L_S$ follows the definition in Equation 14, where $\hat{L}_t$ does not correspond to the target-domain classification loss in Equation 17. The latter is defined as:

$$\hat{\mathcal{L}}_{\mathcal{T}}\left(f_T\left(Z^T\right)\right) = -\frac{1}{N_T}\sum_{i=1}^{N_T} y_i^T \log\left(\hat{y}_i^T\right) \tag{17}$$

In this context, the difference $L_s - \hat{L}_t$ serves as a measure of the relative performance between the source and target domains.

## H  FURTHER ANALYSIS OF SAD

**Proposition 1** (Bound for $D_{S,T}^{\gamma}(P;\lambda)$ (Fang et al., 2025a)). For any $\gamma \geq 0$, and under the assumption that the prior distribution $P$ over the classification function family $\mathcal{H}$ is defined, we establish a bound for the domain discrepancy measure $D_{S,T}^{\gamma/2}(P;\lambda)$. Specifically, we have the following inequality:

$$D_{S,T}^{\gamma/2}(P;\lambda) \leq O\left(\sum_{i\in V^S}\sum_{j\in V^T}||(A^S X^S)_i - (A^T X^T)_j||_2^2 + \sum_{i\in V^S}\sum_{j\in V^T}||X_i^S - X_j^T||_2^2\right). \tag{18}$$

**Experiment Setting:** To assess the cross-domain structural mismatch, we compute two complementary discrepancy measures. First, following the procedure described in the GAA framework Fang et al. (2025a), we evaluate the structural bound value by calculating the first-term deviation $||(A^S X^S)_i - (A^T X^T)_j||_2^2$ as outlined in Appendix F of GAA. This provides a fixed-hop, first-order structural discrepancy between source and target graphs. Second, we report the Structural Aggregation Distance (SAD) used in SAGA, defined in Eq. (3). SAD computes the Frobenius-norm divergence between source and target embeddings after $k$-hop structural aggregation and is dynamically updated during training. For each epoch, SAD is evaluated across all view–hop combinations, and the minimum value is selected as the dominant structural signal. All experiments adopt the same multi-view MGDA protocol as the main results, using the standardized ACM and MAG splits, shared encoder–decoder architecture, and identical training hyperparameters. SAD is recalculated once per epoch using the current embeddings without additional backpropagation cost, ensuring a consistent and fair comparison with the static GAA bound.

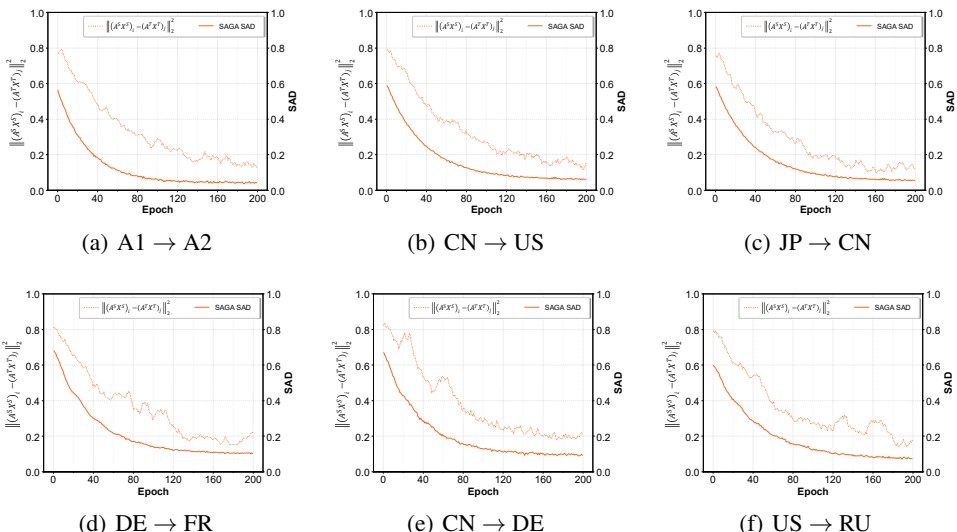

Figure 9: Training dynamics of SAGA across six domain-shift settings. The $x$-axis denotes training epochs; the left $y$-axis reports the squared feature discrepancy $\|(A^S X^S)_i - (A^T X^T)_j\|_2^2$ used in the GAA bound, while the right $y$-axis shows the Structural Aggregation Distance (SAD) defined in Eq. (3). Each subplot contains four trajectories: the GAA structural discrepancy, its smoothed variant, the dynamically updated SAD, and its smoothed version. Across all scenarios, both SAD curves remain consistently and substantially lower than the GAA-based discrepancies, demonstrating that SAD provides a tighter and more stable measure of cross-domain structural mismatch. Moreover, SAD decreases monotonically and converges during training, whereas GAA curves remain high and fluctuate. These results confirm that SAGA's dynamic SAD offers a more reliable alignment signal, leading to superior adaptation performance compared to the fixed-hop GAA discrepancy.

To more comprehensively evaluate the effectiveness of the proposed Structural Aggregation Distance (SAD), we conduct an additional experiment comparing SAD against the structural term used in the GAA bound. The GAA discrepancy measures pairwise deviations of the form $\|(A^S X^S)_i - (A^T X^T)_j\|_2^2$, capturing only the first-hop aggregated mismatch between source and target nodes. Although both SAD and the GAA term quantify how structural aggregation influence node representations, the GAA estimator is fundamentally fixed: it is tied to a predetermined hop depth and cannot update as training progresses. In contrast, SAD is explicitly designed as a dynamic view–hop discrepancy measure that recomputes structural differences across all view–hop combinations and selects the dominant pair adaptively at each epoch. This distinction is essential. As illustrated in Fig. **??**, the dynamic SAD curves monotonically decrease alongside the source–target loss gap $L_s - \hat{L}_t$, while fixed-hop or fixed-view vara single fixed hop aggregation does not govern structural discrepancies in multi-view graphs structural discrepancies in multi-view graphs are not governed by any single hop or single relational view; the most transferable view–hop pair evolves during optimization, and capturing this evolution requires a dynamic metric such as SAD.

Moreover, the experiments reveal a consistent performance gap between the two measures. Because the GAA discrepancy $\|(A^S X^S)_i - (A^T X^T)_j\|_2^2$ relies on a static aggregation operator, its trajectories remain high and fluctuate throughout training. In contrast, SAGA re-estimates SAD at every epoch and updates the dominant view–hop pair $(v^*, K^*)$ accordingly. As shown in Fig. 9, both SAD curves (raw and smoothed) remain substantially lower than the GAA-based curves across all domain-shift settings and exhibit a clear monotonic convergence. This indicates that SAD provides a tighter and more stable estimate of cross-domain structural mismatch and explains why SAGA consistently achieves superior adaptation performance compared to any estimator based on a fixed-hop structural discrepancy.

# I    VISUALIZATION ANALYSIS ON SAGA DYNAMIC SELECTION

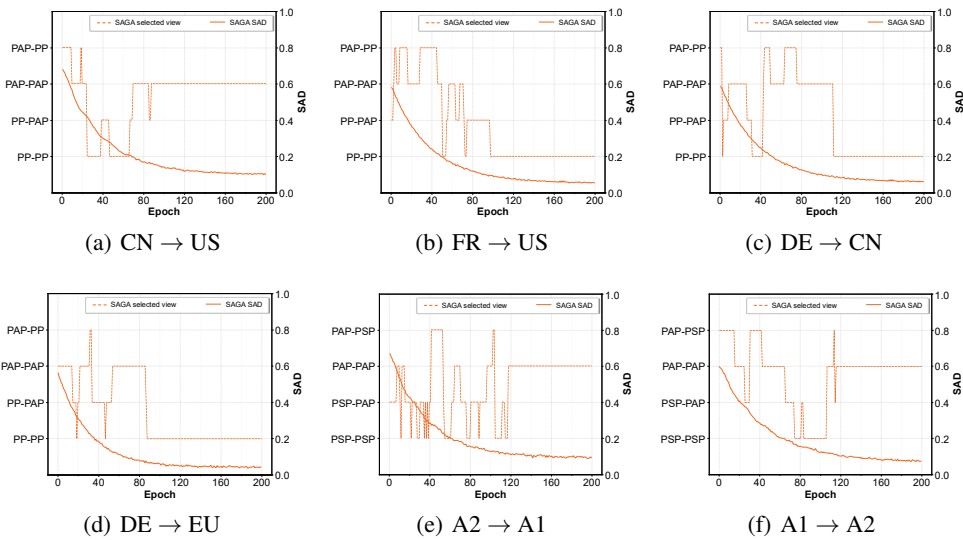

Figure 10: Each subplot illustrates the evolution of the selected principal view–hop pair throughout training. At early stages, SAGA frequently switches among multiple candidate view combinations as it searches for the most structurally compatible alignment configuration. However, as training progresses and the Structural Aggregation Distance (SAD) steadily decreases, the selection pattern stabilizes and consistently converges to a *same view to same view* mapping (e.g., PAP → PAP on ACM, PP → PP on MAG). This behavior reflects the intuitive structural correspondence between source and target graphs and demonstrates that SAGA's dynamic mechanism not only adapts during optimization but also naturally settles into semantically meaningful and stable view alignments.

A core contribution of SAGA lies in its ability to dynamically identify the principal view–order pair that governs cross-domain transferability. Since this mechanism is algorithmically non trivial, we

provide an additional visualization analysis to help reviewers intuitively understand how the dynamic selection behaves during training and why it is necessary. In the visualizations we provide, each training epoch is associated with two trajectories: the Structural Aggregation Distance SAD, and the selected dominant view–hop pair $v^*$. The results show a clear and consistent pattern across both ACM and MAG datasets. First, SAD decreases steadily as training progresses, indicating that SAGA continuously strengthens structural alignment between the source and target graphs. Unlike fixed hop or fixed view variants whose SAD may stagnate or even fluctuate dynamic SAD monotonically declines alongside the reduction during training, reinforcing its role as a faithful indicator of structural shift. Second, the dominant view–hop pair naturally evolves over time. Early in training, the model often explores multiple view combinations (e.g., PAP→PSP or PSP→PAP in ACM; PAP→PP or PP→PAP in MAG). This reflects the fact that different hops and relational views may temporarily provide more transferable structure during early stage representation shaping. As the embeddings become more aligned, however, the optimal pair gradually stabilizes—and importantly, it stabilizes to combinations consistent with intuitive graph semantics. For instance, in ACM, the selection converges to PAP→PAP, the view conveying the strongest author paper relational consistency across domains. In MAG, the selection similarly converges to PP→PP, which captures the most domain invariant citation patterns. This convergence behavior confirms two key insights: Dynamic selection is necessary, because the best structural signal is not fixed and varies throughout optimization. he final dominant view order is interpretable, consistently settling on the most semantically transferable relation in each dataset. Such visual evidence directly validates our design motivation: structural discrepancies in multi-view graphs are governed not by any single fixed hop or view, but by an evolving dominant structural component. SAGA is the first to model this evolution explicitly, and the visualization analysis clearly demonstrates why dynamic selection is essential and how it leads to superior adaptation performance.

