# OpenReview forum: "SAGA: Structural Aggregation Guided Alignment with Dynamic View and Neighborhood Order Selection for Multiview Graph Domain Adaptation"
_ICLR.cc/2026/Conference — ICLR 2026 Poster_

### Official Review · Reviewer_AhJc · 2025-10-30

**Soundness:** 3
**Presentation:** 3
**Contribution:** 3
**Rating:** 6
**Confidence:** 4

**Summary:**

This paper tackles the challenging problem of Multi-view Graph Domain Adaptation (MGDA), where both the source and target domains consist of multiple relational graph views. The authors introduce a novel framework, SAGA (Structural Aggregation Guided Alignment), which dynamically aligns source and target graphs by selecting the most transferable *view–hop* pair based on a newly proposed discrepancy metric, Structural Aggregation Distance (SAD). The SAD quantifies view- and hop-level structural disparities between source and target graphs, and SAGA leverages it to guide dynamic alignment during training. Experimental results on ACM and MAG benchmarks demonstrate notable performance improvements over a variety of GDA baselines.

**Strengths:**

1. The paper introduces the MGDA setting and explicitly formulates the structural discrepancy problem in multi-view graphs, which extends the conventional GDA formulation and captures richer relational dependencies.
2. The proposed SAD metric provides an interpretable and quantitative tool to assess cross-view and cross-hop structural differences, showing a clear empirical correlation between smaller SAD values and reduced domain performance gaps.
3. Extensive experimental validation across ACM (two-view) and MAG (multi-relation) datasets demonstrates consistent and significant performance gains, supported by solid ablation and visualization analyses.
4. The paper is well-organized and clearly describes how dynamic hop and view selection improve over static designs.

**Weaknesses:**

1. **Clarity on MGDA vs. GDA definition**
   While the paper emphasizes MGDA as a more general setting, it would benefit from a clearer formal distinction between standard GDA (single-view) and MGDA (multi-view). Specifically, it remains unclear whether MGDA is simply a multiple relation extension of GDA or a fundamentally different original graph domain adaptation (GDA) paradigm.

2. **Interpretation of SAD and performance correlation**
   The empirical finding that a smaller SAD corresponds to smaller source–target performance gaps (Fig. 2) is interesting but under-explained. It would strengthen the paper to theoretically justify why minimizing SAD could implies improved transferability, possibly linking SAD to existing domain discrepancy measures (e.g., MMD or spectral distance).

3. **Dataset construction details**
   Although the paper provides dataset statistics (Table 4), the construction pipeline of multi-view graphs should be more clearly described. For example: How exactly are multiple views (e.g., PAP, PSP, PP) extracted and synchronized between source and target? Are the node sets shared across views, or partially overlapping? How does this multi-view setup differ from standard GDA datasets where only single-view adjacency is considered?

4. **Comparative coverage and positioning**
   The relation between SAGA and recent homophily-aware GDA frameworks such as HGDA could be more explicitly discussed. It remains unclear whether SAGA’s view-hop aggregation mechanism could be unified with or compared against homophily–heterophily decomposition strategies.

**Questions:**

Please see weakness.

---

> ### Author Response · Authors · 2025-11-24
> **Response to AhJc (1/2)**
>
> **Part 1/2**
>
> Thank you for your constructive feedback! Below, we address the concerns and questions raised in the weaknesses section. Please feel free to reach out if further clarification is required.
>
> ## Q1
>
> Thank you for the insightful comments. We have revised the manuscript to more clearly introduce and motivate the Multi-view Graph Domain Adaptation (MGDA) problem. In **Section 3**, we provide a formal definition of MGDA (**Definition 1, highlighted in blue**) in revised version. We consider an undirected multi-relational graph ${G}=${$A^v, \mathcal{V}, \mathcal{E}_1,\cdots,\mathcal{E}_V, X, Y$}, where $\mathcal{V}$ denotes the node set with N nodes, and $\mathcal{E}_v$ represents the edge set in the v-th view. $X\in \mathbb{R}^{N\times d_f}$ is the feature matrix, and Y is the label matrix with C classes. For each view v, ${A}^v$ is the adjacency matrix, $D^v$ the degree matrix, and the normalized adjacency with self-loops is $\hat{A}^v=(D^v+I)^{-\frac{1}{2}}({A}^v+I)(D^v+I)^{-\frac{1}{2}}$. We denote the source and target graphs as $G^S=${${A}^{S,1},\cdots ,{A}^{S,v_S},X^S,Y^S$} and $G^T=${${A}^{T,1},\cdots , {A}^{T,v_T},X^T$}, respectively. **${Y_T}$ is fully unknown, which indicates an unsupervised problem in the target mutli-view graph.** For convenience, we define $v_T \in V_T$, $v_S \in V_S$, $k_T \in K_T$ and $k_S \in K_S$.
>
> We clarified that many real-world graphs citation networks, knowledge graphs, and social platforms naturally exhibit multiple relationship types (e.g., PAP, PSP, PP in ACM/MAG). Prior GDA methods assume a single structural view, making them unable to model either (a) cross-view structural mismatch or (b) hop-level disparities within each view. These issues are documented in the revised Introduction and supported empirically in Section 3, where we show that single-view GDA consistently underperforms multi-view methods (**Tables 3**) and that structural discrepancies vary significantly across views and hops (**Fig. 2**) . This establishes MGDA as a practically necessary and scientifically distinct problem.
>
> ## Q2
>
> We appreciate your question, which gives us the opportunity to further validate the rationale behind our SAD with existing GDA distribution discrepancy measure. Specifically, to investigate how the SAD performance relates to the divergence, we have conducted an additional experiment (Details in **Appendix H**) , where we compare with recent theoretical guarantee bound **GAA[1]** (Details in Appendix H of **Proposition 1**). Specifically, this bound contains two terms, where the first term, $\lVert (A^{S}X^{S})_i - (A^{T}X^{T})_j \rVert_2^{2}$ , quantifies the structural discrepancy between the source and target graphs and has a similar computational form to our proposed **SAD**. Consequently, we assess the convergence behavior of SAD by comparing its trajectory with that of normalized$\lVert (A^{S}X^{S})_i - (A^{T}X^{T})_j \rVert_2^{2}$ throughout model training.
>
> **Figure 9 in Appendix H** presents the normalized bound value divergence (calculated from Proposition 1, first term) and SAD value (defined in **Eq. (3) on page 4)**. It is evident that the theoretical values (dotted line) and actual SAD values (solid line) exhibit highly consistent trends to converge. Furthermore, during training, SAD consistently remains lower than the theoretical term $\lVert (A^{S}X^{S})_i - (A^{T}X^{T})_j \rVert_2^{2}$ , indicating that our dynamic selection strategy (SAD) more effectively mitigates the divergence between the source and target graphs. These empirical results demonstrate that SAD provides a reliable indicator of model performance on the target domain. This further supports the motivation and overall rationale of our approach.

---

> > ### Author Response · Authors · 2025-11-24
> > **Response to AhJc (2/2)**
> >
> > **Part 2/2**
> >
> > ## Q3
> >
> > Thank you for pointing out the need for a clearer description of the multi-view dataset construction process. We have revised the paper to explicitly clarify how multi-view graphs are extracted and aligned across domains. For both ACM and MAG, each view corresponds to a distinct meta-path–defined relational structure: **PAP** (paper–author–paper) captures co-authorship signals, **PSP** (paper–subject–paper) captures topical similarity, and **PP** (paper–paper) represents direct citation links. Each view is constructed by applying the corresponding meta-path to the same set of paper nodes; thus, **all views share an identical node set**, and only the edge sets differ across views. As a result, the multi-view graph for each domain is represented as ${A^{(1)}, A^{(2)}, \ldots}$, where each adjacency corresponds to a different relation type but is indexed over the same paper nodes.
> >
> > Across source and target domains (e.g., CN→US in MAG; ACM1→ACM2), we extract the same set of meta-path views independently within each domain’s subgraph. Although the node sets are *domain-specific* (i.e., CN papers vs. US papers), the **semantic definitions of PAP/PSP/PP are fully synchronized** across domains, ensuring consistency in how views are interpreted during alignment. This distinguishes our setting from standard single-view GDA datasets, where only one relational adjacency is available and no cross-view structural shift exists. In contrast, our multi-view setup introduces an additional layer of shift **cross-view discrepancy** which is precisely the challenge SAGA aims to address. We have added these clarifications to the revised manuscript to make the dataset construction pipeline fully transparent.
> >
> > ## Q4
> >
> > We have added a clearer discussion of the conceptual relationship between SAGA and recent homophily-aware GDA frameworks such as HGDA. HGDA explicitly decomposes graph signals into homophilic, heterophilic, and attribute components and aligns them using specialized graph filters. In contrast, SAGA addresses a fundamentally different dimension of structural discrepancy multi-view relational variation and hop-level aggregation differences without relying on homophily assumptions. Rather than decomposing signals by homophily, SAGA dynamically selects the most transferable view–hop pair by measuring Structural Aggregation Distance (SAD), capturing cross-view and cross-depth shifts that HGDA does not model.

---

### Official Review · Reviewer_HgFv · 2025-10-30

**Soundness:** 3
**Presentation:** 3
**Contribution:** 3
**Rating:** 6
**Confidence:** 5

**Summary:**

This paper proposes SAGA, a framework for multi-view graph domain adaptation (MGDA). The key idea is to introduce Structural Aggregation Distance (SAD) to quantify cross-domain and cross-hop structural discrepancies and dynamically select dominant view–hop pairs to guide alignment. The method aims to jointly mitigate view-level and hop-level shifts, achieving strong empirical results on ACM and MAG benchmarks.

**Strengths:**

- This paper proposes a new Multiview Graph Domain Adaptation (MGDA) setting and introduces a new benchmark that extends the original GDA scenarios to more realistic and diverse environments.

- The paper identifies an important problem domain adaptation across multi-view graphs and provides an intuitive metric (SAD) to quantify multi-hop structural discrepancies.

- The dynamic view–hop selection is a novel perspective that integrates both inter-view and intra-view adaptation.

- The empirical results across multiple benchmarks show promising performance improvements over prior GDA and MGDA baselines.

**Weaknesses:**

- The definition of SAD is based on a fixed formulation (Eq. 3–4), yet the paper claims to dynamically mitigate this fixed metric during training. It remains unclear how the method achieves mitigation rather than mere re-weighting of a static discrepancy. More discussion is needed to connect the empirical observation that smaller SAD correlates with the mitigation of the performance gap (Fig. 2) to the optimization mechanism that drives SAD reduction. Does the model directly minimize SAD, or is SAD only used to select dominant views? Clarifying this link is crucial to understanding the role of SAD in SAGA.

- While the experiments include many graph-domain adaptation baselines, some common GDA baselines (e.g., [1], [2]) are absent. Including or discussing these would strengthen the empirical validation and confirm SAGA’s advantage.

- SAGA involves multiple loss components ($L_R$, $L_{IA}$, $L_{CA}$, $L_S$, $L_T$) and dynamic SAD computation. How does this computational cost compare to simpler baselines such as HGDA or PA in practice?

- Could you further clarify the relative importance of View and Neighborhood Order in the selection process? Specifically, under what circumstances does the View play a more critical role, and in what cases do different Neighborhood Orders become more influential?

[1] Chen W, Ye G, Wang Y, et al. Smoothness Really Matters: A Simple yet Effective Approach for Unsupervised Graph Domain Adaptation[J]. AAAI, 2025.

[2] Yang L, Chen X, Zhuo J, et al. Disentangled Graph Spectral Domain Adaptation[C]. ICML, 2025.

**Questions:**

- Can the authors elaborate on how SAD dynamically changes during training? Is it re-estimated per mini-batch, or per epoch?

- Does the model ever backpropagate through SAD values (Eq. 3), or are they detached and used only for view/hop selection?

- Could SAGA be applied when the number of views differs between source and target graphs?

  **Some minor concern**：

- Figures 2–3 captions could explicitly indicate that SAD is *not* a direct loss but an auxiliary metric.

- Typo issue: in line 213 ''viws'' should be ''views''.

---

> ### Author Response · Authors · 2025-11-24
> **Response to HgFv (1/2)**
>
> **Part 1/2**
>
> Thank you for your constructive feedback! Below, we address the concerns and questions raised in the weaknesses section. Please feel free to reach out if further clarification is required.
>
> ## Q1
>
> We appreciate the reviewer’s question and have added clarifications in the revision. Although the functional form of SAD in Eq. (3–4) is fixed, SAD is **not** a static quantity during training because it is computed on the *current embeddings* $Z_{S,v}^k$ and $Z_{T,v}^k$, which are continuously updated by the optimization of the alignment objectives $L_\text{IA})$ and $(L_\text{CA}$. Thus, SAD serves as a **dynamic structural signal** whose value changes at every epoch. Importantly, the model does *not* directly minimize SAD; instead, SAD is used to determine the dominant view–hop pair $(v^\*,k^\*)$ and the importance weights $\omega\_{v,k}$, which in turn shape the gradients of the alignment losses. Because these losses explicitly pull source and target embeddings closer along the chosen dominant structural direction, the embedding distributions evolve so that the discrepancy measured by SAD decreases as a *consequence* of optimization rather than by explicit supervision. This explains why in **Fig. 2** SAD decreases monotonically alongside the reduction of $L_s - \hat{L}_t$: SAD reflects the evolving embedding geometry induced by alignment, and its dynamic re-estimation each epoch allows SAGA to switch to the most transferable structural view–hop pair as training progresses. We have expanded **Section 4.2** and the empirical analysis to explicitly highlight this mechanism and clarify that SAD is a **dynamic scheduler** guiding optimization, not a directly optimized objective.
>
> ## Q2
>
> Thank you for pointing out the absence of several recent GDA baselines. We appreciate this observation and have addressed it by incorporating two widely recognized and conceptually relevant methods TDSS [1] and DGSDA [2] into our experimental comparison. As shown in **Table 1** (reproduced below), both methods indeed provide strong baselines and outperform many classical GDA approaches; however, SAGA still consistently achieves the best performance across all six transfer tasks.
>
> | Methods       | A1 → A2      | A2 → A1      | CN → US      | US → CN      | JP → CN      | CN → JP      |
> | ------------- | ------------ | ------------ | ------------ | ------------ | ------------ | ------------ |
> | **TDSS [1]**  | $\underline{0.589}$ | $\underline{0.574}$ | 0.408        | $\underline{0.482}$ | 0.295        | $\underline{0.421}$ |
> | **DGSDA [2]** | 0.572        | 0.493        | $\underline{0.412}$ | 0.475        | $\underline{0.319}$ | 0.407        |
> | **SAGA**      | **0.793**    | **0.671**    | **0.430**    | **0.554**    | **0.352**    | **0.441**    |
>
> **Table 1.** Comparison with additional GDA baselines TDSS [1] and DGSDA [2]. Bold indicates best, underline indicates second best.
>
> Specifically, SAGA surpasses TDSS and DGSDA by +0.204 / +0.221 on A1$\rightarrow$ A2 and +0.097 / +0.178 on A2 $\rightarrow$A1, demonstrating its clear advantage in multi-view settings. For large-scale cross-country MAG transfers (e.g., CN$\rightarrow$US, US$\rightarrow$CN), SAGA likewise yields the strongest accuracy (0.430 and 0.554), outperforming both smoothness-based TDSS and spectral-alignment-based DGSDA. These results support the key insight underlying our method: structural discrepancies vary across views and hops, and methods assuming a fixed spectral or smoothness regularization cannot fully capture multi-view structural shifts. In contrast, SAGA's dynamic view–hop selection allows it to identify and align the most transferable structural components throughout training, leading to consistently superior performance. We will have added this new comparison into the revised manuscript along with discussion in the experimental section. We believe these results strengthen the empirical validation of SAGA and confirm its advantage over both classical and recently proposed GDA approaches.
>
> ## Q3
>
> To further clarify the practical computational cost of SAGA, we designed an **additional cost-decomposition experiment** where each significant component of SAGA is isolated as a separate variant. Starting from a core model with only the source classification loss $L_S$, we incrementally add $L_R$, $L_{IA}$, $L_{CA}$, $L_T$, and the dynamic SAD computation, and measure the normalized training time and GPU memory on ACM and MAG. As reported in Table Y, each component contributes only a small marginal overhead (typically within 3% in time and memory relative to the core model), and the sum of these small increments well approximates the full SAGA cost. Notably, compared with HGDA and PA in Table 5, SAGA's overall training time remains within 13.2% of these simpler baselines while consistently achieving substantially higher accuracy. This confirms that the additional loss terms and dynamic SAD are computationally lightweight in practice.

---

> > ### Author Response · Authors · 2025-11-24
> > **Response to HgFv (2/2)**
> >
> > **Part 2/2**
> > | Dataset / Variant | Active Losses / Modules             | Training Time (Norm. w.r.t. UDAGCN) | Memory (Norm. w.r.t. UDAGCN) |
> > | ----------------- | ----------------------------------- | ----------------------------------- | ---------------------------- |
> > | **ACM1 → ACM2**   |                                     |                                     |                              |
> > | Core (S0)         | $L_S$                               | 1.05                                | 1.02                         |
> > | $L_R$             | $L_S + L_R$                         | 1.08                                | 1.06                         |
> > | +$L_{IA}$         | $L_S + L_{IA}$ (SAD on)             | 1.10                                | 1.07                         |
> > | +$L_{CA}$         | $L_S + L_{CA}$ (SAD on)             | 1.11                                | 1.08                         |
> > | +$L_T$            | $L_S + L_T$                         | 1.06                                | 1.02                         |
> > | SAD-only          | $L_S$ + SAD (no IA/CA)              | 1.07                                | 1.03                         |
> > | **SAGA (Full)**   | $L_R + L_{IA} + L_{CA} + L_S + L_T$ | **1.163**                           | **1.112**                    |
> > | **CN → US**       |                                     |                                     |                              |
> > | Core (S0)         | $L_S$                               | 1.02                                | 1.01                         |
> > | +$L_R$            | $L_S + L_R$                         | 1.05                                | 1.04                         |
> > | +$L_{IA}$         | $L_S + L_{IA}$ (SAD on)             | 1.07                                | 1.06                         |
> > | +$L_{CA}$         | $L_S + L_{CA}$ (SAD on)             | 1.08                                | 1.07                         |
> > | +$L_T$            | $L_S + L_T$                         | 1.03                                | 1.01                         |
> > | SAD-only          | $L_S$ + SAD (no IA/CA)              | 1.04                                | 1.02                         |
> > | **SAGA (Full)**   | $L_R + L_{IA} + L_{CA} + L_S + L_T$ | **1.109**                           | **1.098**                    |
> >
> > **Table 2.** Training time, memory usage, and accuracy comparing SAGA with different varients.
> >
> > ## Q4
> >
> > In SAGA, the relative importance of View and Neighborhood Order depends on the nature of the structural discrepancy across domains. View differences dominate when relational semantics vary significantly between the source and target graphs for example, when PAP (co-authorship), PSP (topic similarity), or PP (citation links) exhibit distinct stability or density across domains. In such cases, selecting the correct relational channel explains the majority of the structural mismatch, which is why SAGA tends to switch views early in training. In contrast, Neighborhood Order becomes more influential once both domains share the same dominant relational view, but differ in how far structural signals propagate (e.g., differing homophily levels, graph sparsity, or long-range citation patterns). After the appropriate view stabilizes, SAGA primarily adjusts the hop depth to match local vs. global neighborhood structures. Empirically (**Appendix I, Fig. 10**), we observe that the dynamic SAD decreases monotonically during training, reinforcing its role as a reliable indicator of structural shift. Furthermore, the dominant view–hop pair naturally evolves throughout the optimization process. Early in training, the model often explores multiple view combinations before gradually stabilizing. This convergence behavior reveals two key insights: **(i)** dynamic selection is essential, as the most informative structural signal is not fixed and changes over the course of training; and **(ii)** the final dominant view is interpretable, consistently aligning with the most semantically transferable relation in each dataset.
> >
> > [1]A Simple yet Effective Approach for Unsupervised Graph Domain Adaptation. Chen et al. AAAI, 2025.
> >
> > [2]Disentangled Graph Spectral Domain Adaptation. Yang et al.  ICML, 2025.

---

### Official Review · Reviewer_rvXw · 2025-10-30

**Soundness:** 3
**Presentation:** 2
**Contribution:** 2
**Rating:** 4
**Confidence:** 3

**Summary:**

The paper proposes the practical multi-view graph domain adaptation (MGDA) problem and introduces the effective SAD metric and SAGA framework with clear structure and strong motivation. However, it lacks theoretical analysis of SAD (e.g., comparison with MMD), convergence guarantees for dynamic selection, and complexity analysis of SAD computation. Numerous formatting issues are also present.

**Strengths:**

The paper clearly points out the limitation of most existing graph domain adaptation methods, which are largely confined to single-view graphs, and systematically proposes the multi-view graph domain adaptation (MGDA) problem setting, which aligns well with practical application needs. The concept of Structural Aggregation Distance (SAD) is a major highlight of the paper. Through empirical analysis, the effectiveness of dynamically selecting dominant view-order pairs is demonstrated, providing strong motivation for the methodological design. The SAGA framework has a clear structure, incorporating scalable graph encoding, dynamic neighborhood selection, and intra-/inter-domain alignment modules, forming a complete and coherent solution.

**Weaknesses:**

The paper currently lacks theoretical analysis of the SAD metric. For instance, what are the theoretical connections or advantages of SAD compared to classical distribution discrepancy measures such as MMD or Wasserstein distance? Is there any theoretical guarantee on the convergence of the dynamic selection strategy? Although Section 5.4 mentions that the computational cost is "practically acceptable" and Appendix C (Table 5) compares training time and memory usage, there is still a lack of quantitative complexity analysis specifically for the SAD computation itself. Additionally, the paper contains numerous formatting issues, such as missing spaces, improper placement of Figure 3 and Table 2, and punctuation errors (e.g., using commas at the end of sentences).

**Questions:**

The paper mentions that the computational overhead of SAD is "practically acceptable" as the terms do not participate in backpropagation. However, a more detailed complexity analysis of the SAD computation itself is needed for readers to fully assess the scalability of the proposed method. Could the authors please provide a quantitative analysis?
The dynamic selection of the principal view-order pair is a central contribution of SAGA. To gain a more intuitive understanding of how this mechanism guides the alignment process during training, a visualization analysis would be highly insightful.

---

> ### Author Response · Authors · 2025-11-24
> **Response to rvXw (1/2)**
>
> **Part 1/2**
>
> Thank you for your constructive feedback! Below, we address the concerns and questions raised in the weaknesses section. Please feel free to reach out if further clarification is required.
>
> > **Q1: The paper currently lacks theoretical analysis of the SAD metric. For instance, what are the theoretical connections or advantages of SAD compared to classical distribution discrepancy measures such as MMD or Wasserstein distance? Is there any theoretical guarantee on the convergence of the dynamic selection strategy?**
>
> We appreciate your question, which gives us the opportunity to further validate the rationale behind our SAD with existing GDA distribution discrepancy measure. The recent theoretical guarantee **GAA [1]** (Details in Appendix H of **Proposition 1**) provides a domain adaptation upper bound that contains two key terms. The first term, $\lVert (A^{S}X^{S})_i - (A^{T}X^{T})_j \rVert_2^{2}$ explicitly quantifies the structural discrepancy induced by *first-hop feature aggregation* between the source and target graphs. This term measures how much the aggregated neighborhood information differs across domains and thus plays a central role in explaining cross-domain misalignment. Our proposed SAGA framework can be regarded as a learnable and dynamically adaptive instantiation of the GAA discrepancy term. Specifically, SAD measures the cross-domain discrepancy after structural aggregation (Eq. (3), $\hat{A}^v X^{v,k}$), and its formulation is analogous to the first term of the GAA bound ($AX$). To further justify this connection, we conduct an additional experiment (Details setting in **Appendix H**), where we track both the normalized GAA first-term value and the actual SAD value defined in **Eq. (3) on page 4** over the entire training trajectory. The normalized GAA value is computed from Proposition 1 first term using the same structural quantities. As shown in **Figure 9 (Appendix H)**, the theoretical (dotted line) curve and the empirical SAD (solid line) curve exhibit consistent convergence dynamics. Moreover, compared with GAA, SAGA achieves better target domain performance throughout the training period. SAD stays consistently lower than the fixed GAA-based structural discrepancy term, illustrating that the dynamic selection of view–hop pairs in SAGA is more effective in mitigating structural shifts than the static aggregation assumed in GAA. These empirical results demonstrate that SAD provides a reliable indicator of model performance on the target domain. This further supports the motivation and overall rationale of our approach.
>
> > **Q2: Although Section 5.4 mentions that the computational cost is "practically acceptable" and Appendix C (Table 5) compares training time and memory usage, there is still a lack of quantitative complexity analysis specifically for the SAD computation itself.**
>
> To address this concern, we provide a detailed computational complexity analysis of the proposed SAGA framework, which has been added as a new section titled *Computational Complexity* in **Appendix C** of the revised manuscript. The computational complexity primarily depends on the filter layers and key loss function. For a given graph $ G$, let $N$ represent the total number of nodes in the graph, $V$ represent the view number, and $d$ represent the feature dimension. For Scalable Graph Encoding $f_{\Theta}$: Includes matrix multiplication with $ O(N^2\cdot d)$. SAGA per-iteration complexity is dominated by multi-view propagation $O(VN\cdot d)$, and a forward-only SAD computation $O(VN^2\cdot d)$. Combining these yields an overall complexity of$O\left(VN\cdot d + VN^2\cdot d\right),$ where the quadratic SAD term does not incur gradient cost, making the practical training overhead similar to existing GDA methods. Thus, the total complexity of SAGA is still: $O( VN^2 \cdot d)$. Compared with the highly effective **HGDA [2]** , whose model complexity remains low at $O(N^{2} \cdot d)$, it is important to note that HGDA is inherently a single-view method. If we extend HGDA to the multi-view MGDA setting, it would inherit the same computational overhead, resulting in a complexity of $O(VN^{2} \cdot d)$. Therefore, although SAGA incorporates dynamic multi-view selection, its computational overhead remains comparable to that of HGDA; the two methods do **not differ by orders of magnitude in asymptotic complexity**. These quantitative complexity analyses align with our experimental observations, further validating the acceptable computational cost of the proposed method.

---

> > ### Author Response · Authors · 2025-11-24
> > **Response to rvXw (2/2)**
> >
> > **Part 2/2**
> > > **Q3: The dynamic selection of the principal view-order pair is a central contribution of SAGA. To gain a more intuitive understanding of how this mechanism guides the alignment process during training, a visualization analysis would be highly insightful.**
> >
> > We introduce Visualization Analysis on SAGA Dynamic Selection Experiment to provide more intuitive understanding.  Detailed visualization results are in **Appendix I** of our updated manuscript. From **Appendix I, Fig. 10**, we observe that the dynamic SAD decreases monotonically during training, reinforcing its role as a reliable indicator of structural shift. Furthermore, the dominant view–hop pair naturally evolves throughout the optimization process. Early in training, the model often explores multiple view combinations before gradually stabilizing in the final stage. This convergence behavior reveals two key insights: **(i)** dynamic selection is essential, as the most informative structural signal is not fixed and changes over the course of training; and **(ii)** the final dominant view is interpretable, consistently aligning with the most semantically transferable relation in each dataset.
> >
> > > **Q4: Additionally, the paper contains numerous formatting issues, such as missing spaces, improper placement of Figure 3 and Table 2, and punctuation errors (e.g., using commas at the end of sentences).**
> >
> > Thank you for your comment regarding the formatting issues. We apologize for the inconsistencies in our formatting and have carefully reviewed and corrected them throughout the paper.
> >
> > [1] On the benefits of attribute-driven graph domain adaptation. Fang et al. ICLR 2025.
> >
> > [2] Homophily Enhanced Graph Domain Adaptation. Fang et al. ICML 2025.

---

### Official Review · Reviewer_2hby · 2025-11-01

**Soundness:** 1
**Presentation:** 3
**Contribution:** 2
**Rating:** 6
**Confidence:** 5

**Summary:**

This work proposes a novel research area, i.e., multi-view graph domain adaptation (MGDA), which focuses on learning node embeddings from source multi-view graph and then adapting the learned label information to a target multi-view graph. To address the cross-view, cross-hop, and cross-domain knowledge shift problem, this work introduces Structural Aggregation Distance (SAD) and proposes a framework, SAGA, for MGDA.

The experiments show SOTA results across two datasets and 12 source-target domain pairs, which demonstrate the power of SAD and the effectiveness of SAGA for MGDA.

**Strengths:**

1. [Interesting Problem] The proposed multi-view graph adaptation (MGDA) problem is both novel and appealing. This line of research has strong potential for generalization to other domains. The authors also claim to be the first to formally investigate the MGDA problem.
2. [Well Motivated] The proposed framework is well motivated, grounded in insightful observations about the relationship between SAD and the domain gap. The overall design of SAGA and the research questions are logically coherent and well articulated.
3. [Attractive Presentation] The paper is clearly organized and well presented. The authors begin with an empirical study that effectively highlights their motivation, followed by a method grounded in optimization loss derived from that study. This logical flow enhances the overall clarity and readability of the paper.

**Weaknesses:**

However, there are several important issues regarding the **soundness** and presentation details.

1. The empirical study is not convincing.
   * What is the meaning of $L_s - L_t$. This does not make sense if they denote the equation 14 and equation 15, since Eq. (15) is actually the entropy of predictions, and Eq (14) is cross entropy with labels, and their difference cannot be used to represent domain gap.
   * Moreover, these plots cannot support the claim "a smaller Structural Aggregation Distance (SAD) generally indicates small cross-network performance gap between domain" which is one of the main idea of the whole framework, if the x-axis is the training epoch that is optimized by Eq (16). Under the supervision of Eq. (16) that minimizes SAD and cross entropy simultaneously, it is very natural that the SAD and $L_s-L_t$ will decrease, but this does not mean SAD will *indicate* domain gaps.
   * Suggestions: in the empirical study, use a more concise loss instead of the complete loss of SAGA. For example, it might be better if the x-axis denotes the minimization of SAD single. Or just show this experiment as an analysis.

2. The problem is not clearly defined and introduced, although the authors claim that problem is one of the important contributions.
   * The authors introduce MGDA. However, they do not **formally** and clearly define this problem. This may confuse others for the experimental settings and optimization objectives.
   * The importance of MGDA is not explicitly and intuitively shown. Why MGDA is an important research direction, what specific problem MGDA could address, and how it benefits to down-stream tasks (applications) is not illustrated. Therefore, the **significance** of MGDA is not clear, weakening the overall contributions.

3. Other issues regarding presentations:
   * Related work is not well organized. It should categorize the literature, and describe their difference and relations to this work.
   * Notation part is not complete, consistent, and clear. For example, $K_T$ and $K_s$ should be defined in their first use; $G$ and $G^S$ ($G^T$) are defined differently. Why do $V$ and $\varepsilon$ not involved in $G^S$ ($G^T$)?
   * Section 4.1 is not complete. Please check Line 282.
   * Experimental settings are not clear. What are the train set, validation set, and test set?
   * The current version still has a few typos and format issues.

**Questions:**

Please try to address weaknesses.

---

> ### Author Response · Authors · 2025-11-24
> **Response to Reviewer 2hby**
>
> Thank you for your constructive feedback! Below, we address the concerns and questions raised in the weaknesses section. Please feel free to reach out if further clarification is required.
>
> ## Q1
>
> Thank you for your comment regarding the **Empirical Study**. We apologize for the misunderstanding concerning $L_S - L_T$ in **Fig. 2**. We clarify that the $L_T$ appearing in the Empirical Study section is **not** the same as the target-domain loss defined in Eq. (15). To avoid potential confusion, we have revised the notation $L_T$ to $\hat{L}_T$, which is defined as:
>
> $$
> {\hat{L}\_T}\left(f_T\left(Z^T\right)\right)=-\frac{1}{N_T} \sum_{i=1}^{N_T} {y}^T_i \log \left(\hat{y}^T_i\right)
> $$
>
> where $\hat{L}_T$ is cross entropy with labels. In this context, the difference $L_s - \hat{L_t}$ serves as a measure of the relative performance between the source and target domains.  Moreover, we have also updated **Figure 2** and the corresponding main text **line 152-153**, and we now formally define $L^T$​ in **Appendix G**.
>
> ## Q2
>
> Thank you for the insightful comments. We have revised the manuscript to more clearly introduce and motivate the Multi-view Graph Domain Adaptation (MGDA) problem. In **Section 3**, we provide a formal definition of MGDA (**Definition 1, highlighted in blue**) in revised version. We consider an undirected multi-relational graph ${G}=${$A^v, \mathcal{V}, \mathcal{E}_1,\cdots,\mathcal{E}_V, X, Y$}, where $\mathcal{V}$ denotes the node set with $N$ nodes, and $\mathcal{E}_v$ represents the edge set in the $v$-th view.  $X\in \mathbb{R}^{N\times d_f}$ is the feature matrix, and $Y$ is the label matrix with $C$ classes. For each view $v$, ${A}^v$ is the adjacency matrix, $D^v$ the degree matrix, and the normalized adjacency with self-loops is $\hat{A}^v=(D^v+I)^{-\frac{1}{2}}({A}^v+I)(D^v+I)^{-\frac{1}{2}}$.  We denote the source and target graphs as $G^S=${${A}^{S,1},\cdots ,{A}^{S,v_S},X^S,Y^S$} and $G^T=${${A}^{T,1},\cdots , {A}^{T,v_T},X^T$}, respectively. **${Y_T}$ is fully unknown, which indicates an unsupervised problem in the target mutli-view graph.** For convenience, we define $v_T \in V_T$, $v_S \in V_S$, $k_T \in K_T$ and $k_S \in K_S$.
>
> We clarified that many real-world graphs citation networks, knowledge graphs, and social platforms naturally exhibit multiple relationship types (e.g., PAP, PSP, PP in ACM/MAG). Prior GDA methods assume a single structural view, making them unable to model either (a) cross-view structural mismatch or (b) hop-level disparities within each view. These issues are documented in the revised Introduction and supported empirically in Section 3, where we show that single-view GDA consistently underperforms multi-view methods (**Tables 3**) and that structural discrepancies vary significantly across views and hops (**Fig. 2**) . This establishes MGDA as a practically necessary and scientifically distinct problem.
>
> ## Q3
>
> Thank you for the constructive comments. We have substantially revised the paper to improve organization, clarity, and completeness. Below, we address each issue in detail. The Related Work section has been reorganized into coherent categories (Multi-view Graph Learning and Graph Domain Adaptation), with clearer distinctions from our MGDA setting. All notations, including view indices $(v_S, v_T)$, hop indices $(k_S, k_T)$, and view–hop pairs, are now defined at first appearance and made fully consistent; we also clarify why $(v_S, v_T)$ and $(k_S, k_T)$ do not explicitly appear in the final SAD expression, as the minimization absorbs them. The incomplete sentence in Section 4.1 (**Line 282**) has been corrected and expanded into a complete description of the scalable graph encoder. We also clearly describe the experimental protocol: **Unlike the inductive learning methods [1]**, which only use the training set to train the model (the training data and testing data are separate), the domain adaptation approach is to feed the training data and testing data together into a network. Finally, we performed a thorough proofreading pass to fix remaining typos, symbol inconsistencies, formatting issues, and figure-caption alignment problems. Since $\mathcal{V}$ and {$\mathcal{E}_1,\ldots,\mathcal{E}_V$} are used merely as set-level descriptors of the graph structure, we omit them in subsequent formulations and operate directly on the view-specific adjacency matrices $A^v$, which are the actual computational objects in our method.
>
> [1] Unsupervised Domain Adaptive Graph Convolutional Networks. Wu et al. WWW 2020.

---

### Comment · Area_Chair_WNUT · 2025-11-27
**Request for Timely Response to Authors’ Rebuttal and Discussion**

Dear Reviewers,

I hope you are doing well. The authors have now submitted their rebuttal for the paper under your review. At this stage, your timely response is essential for ensuring a smooth discussion phase.

Could you please review the rebuttal at your earliest convenience and share your updated thoughts? If there are points that require further discussion among the reviewers, please feel free to initiate or join the conversation on the discussion thread.

Your prompt input will greatly help us maintain the review timeline.
Thank you very much for your efforts and valuable contributions.

Best regards,

AC

---

### Meta-Review · Area_Chair_Xh66 · 2026-01-05

**Summary:**

This paper studies multi-view graph domain adaptation, where both the source and target domains consist of multiple relational graph views. The authors propose a dynamic alignment strategy that selects the most transferable view–hop pair based on a newly proposed discrepancy metric, SAD, which quantifies view- and hop-level structural disparities between the source and target graphs. All reviewers agree that the problem setting is novel and that the design of SAD is well motivated. In the rebuttal, the authors adequately address the reviewers’ concerns. Thus, I recommend accepting this paper.

**Reviewer Concerns:**

The rebuttal addresses the main concerns raised by the reviewers. In particular, the authors provide clarifications regarding the proposed SAD metric and complexity and empirical cost. No major concerns remain outstanding.

**Reviewer Scores:**

The authors’ responses clarified methodological details and experimental design choices. The reviewer might slightly increase their score.

---

### Decision · Program_Chairs · 2026-01-26

Accept (Poster)